# SCHEMIXQA AND CORE-VLM: A BENCHMARK AND COLLABORATIVE REFINEMENT (CORE) FRAMEWORK FOR VISUAL QUESTION ANSWERING ON TECHNICAL SCHEMATICS

## ABSTRACT

We present **SchemixQA**, a multimodal benchmark for evaluating Vision–Language Models (VLMs) on *Visual Question Answering (VQA)* over technical schematics. Unlike previous VQA datasets focused on natural images, SchemixQA targets structured domains such as circuits, flowcharts, logic gates, P&I diagrams, and state diagrams, each paired with natural language questions and multiple reference answers. To address this setting, we introduce **CoRe-VLM** (Collaborative Refinement for VLMs), the first actor–critic inspired refinement framework for schematic VQA. In CoRe-VLM, an actor VLM generates answers, while a critic VLM verifies them and provides corrective feedback. A fallback mechanism ensures robustness by reverting to the actor's output when the critic introduces errors. We benchmark seven state-of-the-art VLMs, including GPT-4o, Gemini, Qwen2 and LLaVA, under single-pass and CoRe-VLM inference. The results show that CoRe-VLM consistently improves lexical (Exact Match, BLEU, ROUGE-L) and semantic (BERTScore, Macro/Micro-F1) metrics, with especially strong gains for weaker open source actors when paired with a strong critic. Together, SchemixQA and CoRe-VLM establish a new foundation for domain-specific multimodal reasoning.

## 1 INTRODUCTION

Recent advancements in large-scale vision-language models (VLMs) have significantly enhanced multimodal reasoning capabilities across natural images, charts, and documents Radford et al. (2021); Li et al. (2023); Chen et al. (2023). However, most benchmarks used to evaluate these models have been limited to natural or general-purpose imagery (e.g., COCO, Visual Genome) and domain-agnostic question answering (e.g., VQAv2 Goyal et al. (2017), GQA Hudson and Manning (2019)). In contrast, real-world technical reasoning often requires interpreting structured diagrams such as electrical schematics, digital circuits, signal flows, and logical blocks. These domains pose unique challenges for both visual recognition and semantic interpretation due to symbolic complexity, domain-specific notations, and non-natural language semantics.

To bridge this gap, we introduce **SchemixQA**, a new multimodal benchmark for evaluating Visual Question Answering (VQA) in the context of technical schematic diagrams. The dataset consists of diverse circuit-related illustrations spanning analog and digital domains, including logic gates, combinational and sequential circuits, PID controllers, flowcharts, and state diagrams. Each image is paired with five natural language questions and three human-verified reference answers, enabling fine-grained evaluation of VLM performance on schematic reasoning tasks.

Schematic diagram understanding differs fundamentally from natural image QA. Instead of recognizing objects and scenes, models must identify nodes with symbolic labels (e.g., U1, OUT), interpret arrows and structural layouts, and reason about hierarchical flow. The questions probe taxonomy (e.g., "What gate is used?"), function ("What is the output behavior?"), and topology ("How are inputs connected to the output?"), requiring joint visual and semantic reasoning.

To assess these challenges, we benchmark multiple open-source and commercial VLMs, including Qwen-VL Bai et al. (2024), MiniGPT-4 Zhu et al. (2023), GPT-4o OpenAI (2024), CogVLM Wang et al. (2023), and LLaVA variants Liu et al. (2023). We adopt a unified evaluation framework incorporating Exact Match, BLEU, ROUGE-L, BERTScore, and macro/micro-F1, which capture both lexical accuracy and semantic fidelity.

Beyond benchmarking, we introduce **CoRe-VLM** (*Collaborative Refinement for VLMs*), a novel actor–critic inspired pipeline for schematic VQA. In this framework, an actor VLM generates candidate answers, and a critic VLM re-evaluates the image, question, and actor output to refine or correct the response. This collaborative mechanism leads to substantial improvements in prediction quality compared to single-pass VLM inference.

**Our key contributions are as follows:**

- We introduce **SchemixQA**, the first multimodal benchmark for schematic VQA, covering structured domains such as circuits, flowcharts, logic gates, PID controllers, and state diagrams, each paired with natural language questions and multiple reference answers.

- We establish a rigorous benchmarking protocol by evaluating seven state-of-the-art VLMs (GPT-4o, Gemini, Qwen2, LLaVA) under standardized metrics, including Exact Match, BLEU, ROUGE-L, BERTScore, and macro/micro-F1, providing the first systematic comparison on this domain.

- We propose **CoRe-VLM**, a novel Collaborative Refinement framework that employs an actor–critic style interaction with a fallback mechanism, ensuring robustness while consistently improving both lexical and semantic metrics over single-pass baselines.

- We release **SchemixQA**, the full benchmark results, and an evaluation toolkit to foster reproducibility and future research in domain-specific multimodal reasoning.

We believe that **SchemixQA**, together with the **CoRe-VLM** pipeline, establishes a rigorous foundation for advancing structured visual reasoning, and represents a step toward general-purpose VLMs capable of domain-specific technical understanding.

## 2 RELATED WORK

### 2.1 VQA AND DIAGRAM UNDERSTANDING

Seminal VQA datasets such as COCO-VQA Antol et al. (2015), GQA Hudson and Manning (2019), and CLEVR Johnson et al. (2017) have advanced research in image-based visual question answering. These benchmarks primarily focus on natural scenes or synthetic 3D shapes, testing visual grounding, attribute comparison, and counting. Models such as Neural Module Networks (NMN) Hu et al. (2017) and MAC Hudson and Manning (2018) demonstrate compositional reasoning over visual and textual input. However, these datasets lack the symbolic, diagrammatic, and topological properties central to schematic understanding.

To address diagram-centric reasoning, IconQA Lu et al. (2021) introduced visual QA tasks over science diagrams commonly found in textbooks. AI2D-RST Hiippala et al. (2019) offers richer semantic annotations using rhetorical structures and visual grouping. Diagram Question Generation (DiagramQG) Liu et al. (2024) focuses on automatically generating QA pairs for educational diagrams. Yet, most of these efforts do not incorporate formal graph structures or domain-specific logic like circuit schematics.

Kembhavi et al. Kembhavi et al. (2016) proposed the notion of diagram parse graphs to represent object-relationship semantics in science visuals. Their dataset and model emphasized spatial layout and object connectivity, setting the stage for structure-aware diagram interpretation. Nevertheless, these approaches remain largely disconnected from electrical or digital circuit contexts where symbolic notation and flow semantics are prominent.

## 2.2 CIRCUIT AND SCHEMATIC VQA

VQA over electrical schematics introduces distinct challenges including symbol disambiguation, analog/digital reasoning, and hierarchical flow interpretation. CircuitVQA Mehta et al. (2024) presents a dataset of 5,725 circuit diagrams with over 115,000 QA pairs spanning identification, topology, and spatial reasoning. Their task formulation emphasizes image-level understanding, yet excludes graph-level structure or edge extraction.

Similarly, ElectroVizQA Meshram et al. (2024) focuses on schematic QA in digital logic circuits with a curated dataset targeting undergraduate education. While valuable for logic comprehension, it lacks multimodal diversity (e.g., analog schematics, hand-drawn diagrams) and omits structural representation.

Other prior works in related domains include SKILL Luo et al. (2021), a dataset of scientific knowledge illustrated through diagrams and charts, and ChartQA Masry et al. (2022), which explores concept-level question answering over instructional figures. While these datasets include visually rich sources, none focus on circuit-style visual semantics or symbolic node connectivity.

## 2.3 STRUCTURAL GRAPH EXTRACTION

Most VQA benchmarks are optimized for textual answer prediction. However, technical diagrams such as circuits inherently encode information as graphs: nodes (components) and edges (connections). While Kembhavi et al. Kembhavi et al. (2016) highlighted the importance of extracting diagram parse graphs, no existing benchmarks integrate graph prediction directly into VQA-style tasks.

Research in information extraction from figures (e.g., chart-to-text Kantharaj and Goyal (2023)) or layout-aware document parsing (e.g., PubLayNet Zhong et al. (2019)) is orthogonal in focus. Efforts in visual programming language understanding, such as in block-based code environments, also explore structured extraction, but are unrelated to electrical semantics.

SchemixQA fills this critical gap by introducing natural language question answering over schematic diagrams. It is the first benchmark to specifically evaluate multimodal reasoning on technical circuit illustrations, enabling a rigorous assessment of VLM capabilities in structured, domain-specific visual understanding.

# 3 SCHEMIXQA DATASET

The **SchemixQA** dataset was developed to evaluate Visual Question Answering (VQA) in the context of technical schematic diagrams. Unlike existing VQA benchmarks that focus primarily on natural imagery or general-purpose visual reasoning, SchemixQA targets structured technical domains such as circuit schematics, flowcharts, logic gates, PID controllers, state diagrams, and related symbolic notations.

We curated schematic diagrams from diverse open-source educational repositories, electronic design resources, industrial process documentation, and public-domain collections Kafle et al. (2018); Mendonça et al. (2020); Lu et al. (2022). Each image is paired with a set of natural language questions that probe recognition, functionality, and symbolic reasoning. To ensure linguistic diversity and robustness, every question is annotated with three human-verified reference answers.

The dataset is organized into training and testing splits, supporting standardized benchmarking. In total, SchemixQA contains 500 images, 3,906 questions, and 11,718 answers. Table 1 summarizes the dataset composition.

Table 1: Statistics of the SchemixQA dataset. Each question is paired with three reference answers.

| Split | Images | Questions | Answers |
|-------|--------|-----------|---------|
| Train | 360 | 2,764 | 8,292 |
| Test | 140 | 1,142 | 3,426 |
| **Total** | **500** | **3,906** | **11,718** |

By integrating schematic diagrams with linguistically diverse VQA annotations, **SchemixQA** establishes a rigorous benchmark for structured multimodal reasoning in domain-specific contexts.

Beyond dataset scale, we also stratify questions by type to capture complementary reasoning challenges. Since schematic categories vary in semantics and structure, we define distinct question-type taxonomies for every category of images, which can be depicted in Table 2.

Table 2: Question categories per schematic type in SchemixQA. Each abbreviation is expanded for clarity.

| Category | Question Types |
|---|---|
| Flowcharts | PI (Process Initialization), EH (Exception Handling), DA (Decision Analysis), DI (Document Identification), FD (Flow Destination), SA (Structural Analysis) |
| Logic Gates | GI (Gate Identification), IL (Input Labels), OL (Output Labels), ZCB (Zero-Case Behaviour), SCB (Single-Case Behaviour), TTA (Truth Table Analysis) |
| Circuit Diagrams | CCR (Component Count – Resistors), CCVS (Component Count – Voltage Sources), RL (Resistor Labels), VSL (Voltage Source Labels), CT (Connection Type), CC (Component Calculation), VSB (Voltage Source Behaviour) |
| Valves and P&ID | PI (Pump Identification), CC (Connectivity Check), CD (Connection Details), MEC (Major Equipment Count), PR (Positional Relationship), IC (Instrumentation Count), EP (Equipment Presence) |
| Schematic Diagrams | PS (Protection & Safety), CCT (Core Controller & Timing), PR (Power & Regulation), SA (Sensors & Actuators), IUC (Interfaces & User Controls), I (Indicators) |
| State Diagrams | SE (Structure & Elements), SES (Start & End States), TF (Transitions & Flow), EEH (Error & Exception Handling), KFA (Key Final Actions) |
| PID Controllers | CC (Control Components), MI (Measurement Instruments), C (Controllers), AO (Actuation & Operation), PM (Position Monitoring), SI (System Interaction) |

This structured stratification ensures that **SchemixQA** does not simply measure surface-level textual matching, but instead evaluates a wide spectrum of reasoning skills ranging from symbolic recognition to functional and structural analysis. For clarity, representative visual examples of all question type categories across schematic domains are included in Appendix A.

## 3.1 TASK FORMULATION

The primary task in **SchemixQA** is *Visual Question Answering (VQA)* over technical schematic diagrams. Given an image $I$ and a natural language question $Q$, the model must generate an answer $\hat{A}$ consistent with one of the reference answers $\{A_1, A_2, A_3\}$. Unlike natural-image VQA, schematics demand symbolic recognition, functional interpretation, and structural reasoning.

To capture these skills, we stratify questions by schematic type. **Flowchart** questions probe initialization, exception handling, branching, documentation, flow termination, and structural analysis. **Logic gate** questions address gate identification, input–output labeling, and truth-table reasoning. **Circuit diagram** questions involve component recognition, connectivity, and sub-circuit functionality. **PID controller** questions focus on control types, roles of terms, and system stability. **State diagram** questions examine states, transitions, and terminal conditions. **Schematic block diagram** questions require symbolic label interpretation and hierarchical tracing. Finally, **valves and P&ID** questions target process-control reasoning such as valve types, flow logic, and operational outcomes.

This taxonomy ensures that SchemixQA evaluates not just surface-level answers but the **multi-faceted reasoning** needed for structured technical understanding.

## 3.2 DATA COLLECTION AND ANNOTATION

To build the **SchemixQA** dataset, we collected 500 schematic diagrams spanning circuits, flowcharts, logic gates, PID controllers, state diagrams, schematic block diagrams, and valve-based P&ID processes. These images were sourced from a combination of open educational repositories, electronics resources, and publicly available industrial documentation, ensuring a broad coverage of schematic representations.

To ensure a rich and open-ended question–answer format, we adopted a hybrid construction pipeline. Initially, powerful GPT-5 models were employed to generate candidate VQA pairs for each image, encouraging lexical diversity and semantic variety across the dataset. At the same time, we carefully structured question design along multiple reasoning dimensions, ensuring that each schematic type was annotated with consistent task-specific queries. This balance between automated generation and structured design enabled both coverage and consistency.

Human annotators with technical expertise then reviewed and refined the data. Each question–answer pair was verified for factual correctness, domain appropriateness, and linguistic clarity. Ambiguous or inconsistent outputs were corrected, while redundant phrasing was consolidated. This multi-stage process preserved the diversity of automatically generated annotations while guaranteeing human-level accuracy.

The distribution of images across categories is shown in Figure 1, illustrating that the dataset maintains strong balance across the six primary schematic types, with smaller but meaningful contributions from specialized industrial diagrams. This structured collection and verification process ensures that **SchemixQA** provides both scale and reliability for benchmarking multimodal reasoning.

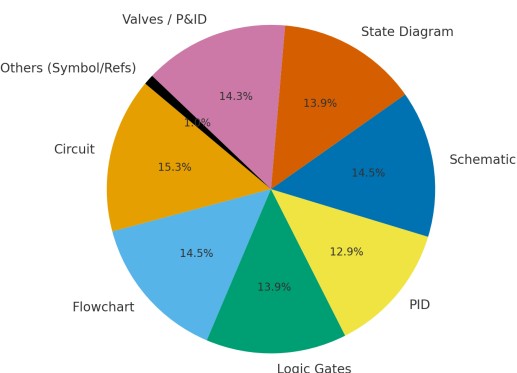

Figure 1: Category distribution in the SchemixQA dataset (500 images).

Further details regarding the annotation process and construction methodology are provided in Appendix B.

## 4 Proposed Method: CoRe-VLM (Collaborative Refinement for VQA)

We propose **CoRe-VLM**, a collaborative refinement framework that improves schematic VQA through an actor–critic inspired pipeline. Given an image $I$ and question $Q$, an *actor* VLM $f_\theta$ produces an initial answer $a_0$. A *critic* VLM $g_\phi$ then re-reads $(I, Q)$ alongside $a_t$ to (i) diagnose potential errors, (ii) provide corrective feedback $r_t$, and (iii) assign a confidence score $s_t \in [0, 1]$. The actor incorporates $r_t$ to generate a refined answer $a_{t+1}$. This process repeats for up to $K$ refinement rounds or until $s_t \geq \tau$. Finally, an aggregator $h(\cdot)$ selects the best answer $\hat{A}$ from $\{a_0, \ldots, a_T\}$ using critic scores and optional self-consistency checks.

**Design principles.** CoRe-VLM follows four principles: (1) *separation of roles*, where the actor focuses on fluent answer generation and the critic focuses on verification; (2) *targeted feedback*, where the critic provides concise, actionable corrections; (3) *budgeted refinement*, where $K$ and $\tau$ control inference cost; and (4) *model-agnostic design*, allowing $f_\theta$ and $g_\phi$ to be distinct models or the same model prompted differently.

**Scoring and stopping.** At round $t$, the critic evaluates symbolic grounding (e.g., circuit components, flowchart nodes), logical consistency, and language quality. The process terminates if $s_t \geq \tau$, if refinements converge ($a_{t+1} \approx a_t$), or when $t = K$. The final output is $\hat{A} = \arg\max_{a_t} s_t$.

### 4.1 Algorithm

We summarize CoRe-VLM as an inference-time algorithm. It does not require retraining and can be applied as a wrapper around existing VLMs. The two key hyperparameters are the maximum number of refinement rounds $K$ and the confidence threshold $\tau$.

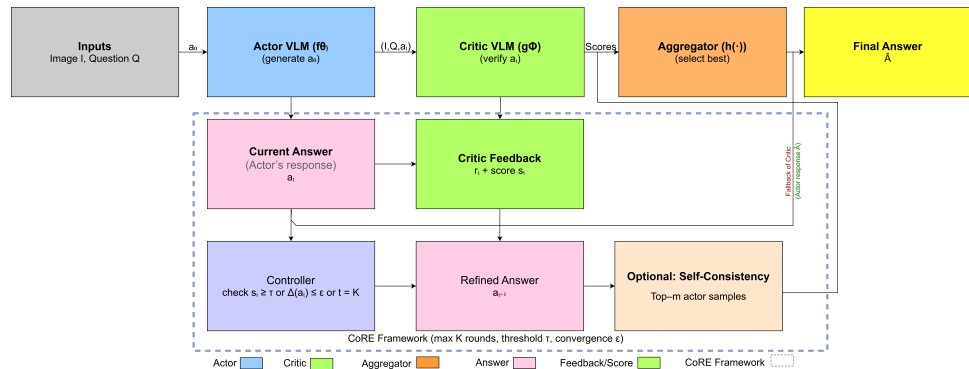

Figure 2: Block diagram of the **CoRe-VLM** pipeline showing actor generation, critic verification and feedback, iterative refinement, and final answer selection.

---

**Algorithm 1** CoRe-VLM: Collaborative Refinement for SchemixQA

---

**Require:** Image $I$, Question $Q$, Actor $f_\theta$, Critic $g_\phi$, budget $K$, threshold $\tau$
**Ensure:** Final answer $\hat{A}$
1: Initialize answer list $A \leftarrow [\,]$
2: Generate initial answer $a_0 \leftarrow f_\theta(I, Q)$
3: Append $(a_0, 0)$ to $A$
4: **for** $t = 0$ to $K$ **do**
5:      Obtain critic feedback and score: $(r_t, s_t) \leftarrow g_\phi(I, Q, a_t)$
6:      Update $A[-1] \leftarrow (a_t, s_t)$
7:      **if** $s_t \geq \tau$ **then**
8:          **break**
9:      **end if**
10:     Generate refinement $a_{t+1} \leftarrow f_\theta(I, Q, \text{feedback} = r_t)$
11:     **if** $\text{EditDist}(a_{t+1}, a_t) \leq \epsilon$ **then**
12:         **break**
13:     **end if**
14:     Append $(a_{t+1}, 0)$ to $A$
15: **end for**
16: Select final answer $\hat{A} \leftarrow \arg\max_{(a,s)\in A} s$
17: **return** $\hat{A}$

---

**Practical notes.** Critic prompts are designed as verification roles (e.g., "Check if the actor's answer matches the diagram and question. Identify errors and provide corrections."). Feedback $r_t$ is kept concise to avoid over-generation. Optionally, self-consistency can be applied by sampling multiple actor outputs per round and selecting the one with highest critic score. Latency grows linearly with $K$, but in practice $K = 1$ or $2$ provides most of the gains with minimal overhead.

## 5 EXPERIMENTAL RESULTS AND DISCUSSION

### 5.1 BENCHMARKING ON SCHEMIXQA

We first benchmarked seven state-of-the-art vision–language models (VLMs) on the **SchemixQA** dataset: GPT-4o, Gemini Pro 1.5, Gemini 1.5 Flash, Gemini 2.0 Flash Lite, Qwen2 VL-Chat 7B, Qwen2.5-VL-7B-Instruct, and LLaVA-1.5-7B. Each model was evaluated in a *single-pass inference* setting without refinement.

Performance was measured using multiple lexical and semantic evaluation metrics:

**Exact Match (EM).** Exact Match directly evaluates whether a predicted answer $\hat{A}$ exactly matches any reference answer $A \in \mathcal{G}$. Formally,

$$\text{EM} = \frac{1}{N} \sum_{i=1}^{N} \mathbb{1}\!\!\!\mathbb{1}\left[\, \hat{A}_i \in \mathcal{G}_i \,\right], \tag{1}$$

where $\mathbb{1}\!\!\!\mathbb{1}[\cdot]$ denotes the indicator function.

**BLEU-1/4 Papineni et al. (2002).** BLEU measures $n$-gram overlap between prediction and reference, penalizing overly short outputs. For $n$-gram precision $p_n$,

$$\text{BLEU} = BP \cdot \exp\left(\sum_{n=1}^{N} w_n \log p_n\right), \tag{2}$$

where $BP$ is the brevity penalty and $w_n$ are weights (commonly uniform). We report BLEU-1 (unigram) and BLEU-4 (up to 4-grams).

**ROUGE-L Lin (2004).** ROUGE-L computes the longest common subsequence (LCS) between prediction and reference, capturing fluency and recall. Given precision $P$, recall $R$, and $\beta$,

$$\text{ROUGE-L} = \frac{(1 + \beta^2) \cdot P \cdot R}{R + \beta^2 P}. \tag{3}$$

**BERTScore Zhang et al. (2020).** BERTScore evaluates semantic similarity using contextual embeddings from pretrained transformers. For predicted token embeddings $\hat{h}_i$ and reference embeddings $h_j$,

$$\text{BERTScore-Precision} = \frac{1}{|\hat{A}|} \sum_i \max_j \cos(\hat{h}_i, h_j), \tag{4}$$

$$\text{BERTScore-Recall} = \frac{1}{|A|} \sum_j \max_i \cos(\hat{h}_i, h_j), \tag{5}$$

and the F1 score is the harmonic mean of the two.

**Macro/Micro F1 Powers (2011).** F1 balances precision and recall. For a class $c$,

$$F1_c = \frac{2 \cdot P_c \cdot R_c}{P_c + R_c}. \tag{6}$$

*Macro-F1* averages $F1_c$ across classes, while *Micro-F1* computes precision and recall globally before applying the F1 formula.

Across these metrics, **Gemini 2.0 Flash Lite** consistently achieved the strongest results in terms of ROUGE-L (0.3781), BERTScore (F1 = 0.3921), and Macro F1 (0.4272). These metrics are particularly important for schematic VQA, as ROUGE-L captures the longest common subsequence and thus reflects structural alignment with reference answers, while BERTScore leverages contextual embeddings to measure semantic fidelity beyond lexical overlap. The higher Macro F1 further indicates that Gemini 2.0 Flash Lite provides more balanced performance across diverse categories, rather than overfitting to frequent patterns.

On the other hand, **Qwen2.5-VL-7B-Instruct** demonstrated superior performance on Exact Match (0.2863), BLEU-1 (0.3514), and BLEU-4 (0.0893). These metrics emphasize exact lexical overlap and local $n$-gram precision, suggesting that Qwen2.5-VL-7B-Instruct is particularly effective when concise symbolic or terminology-level correctness is required.

Together, these results highlight complementary strengths: Gemini 2.0 Flash Lite provides higher semantic and structural fidelity, while Qwen2.5-VL-7B-Instruct excels in lexical precision and exact answer matching. Table 3 reports the detailed benchmarking results. This motivated our choice of Gemini 2.0 Flash Lite as the critic in the CoRe-VLM framework

Table 3: Benchmark results on the SchemixQA dataset (single-pass VLM inference)

| Model | Exact Match | BLEU-1 | BLEU-4 | ROUGE-L | BERTScore (F1) | Macro F1 |
|---|---|---|---|---|---|---|
| GPT-5 | 0.1874 | 0.1385 | 0.0284 | 0.3182 | 0.2032 | 0.3283 |
| Gemini-2.5-Pro | 0.0000 | 0.0212 | 0.0038 | 0.1044 | 0.0500 | 0.1092 |
| GPT-4o | 0.0525 | 0.1812 | 0.0525 | 0.3610 | 0.3809 | 0.3982 |
| Gemini Pro 1.5 | 0.0009 | 0.1285 | 0.0348 | 0.2790 | 0.2866 | 0.3040 |
| Gemini 1.5 Flash | 0.0035 | 0.1499 | 0.0423 | 0.3141 | 0.3244 | 0.3452 |
| Gemini 2.0 Flash Lite | 0.0621 | 0.1671 | 0.0421 | **0.3781** | **0.3921** | **0.4272** |
| Qwen2 VL-Chat 7B | 0.0350 | 0.0629 | 0.0142 | 0.1572 | 0.1882 | 0.1536 |
| Qwen2.5-VL-7B-Instruct | **0.2863** | **0.3514** | **0.0893** | 0.3141 | 0.3531 | 0.3186 |
| LLaVA-1.5-7B | 0.1086 | 0.2703 | 0.0401 | 0.2585 | 0.3692 | 0.2539 |

## 5.2 CoRe-VLM Evaluation

Based on the benchmarking analysis, we adopted **Gemini 2.0 Flash Lite** as the critic in our proposed CoRe-VLM pipeline. Each of the other VLMs was then used as the actor to test whether collaborative refinement improves answer quality.

A fallback mechanism was also integrated: whenever the critic's correction was deemed less reliable than the actor's original answer, the pipeline defaulted to the actor's output. This ensured stability while maintaining refinement gains.

Table 4 presents an example of CoRe-VLM performance when **Qwen2.5-VL-7B-Instruct** is used as the actor and **Gemini 2.0 Flash Lite** serves as the critic. Results show clear improvements across all metrics compared to the single-pass baseline, validating the effectiveness of the actor–critic collaborative refinement approach.

Table 4: Performance of the **CoRe-VLM Framework** on SchemixQA with Gemini 2.0 Flash Lite as the Critic. Each row corresponds to a different Actor model used inside the framework.

| Actor Model | CoRe-VLM Framework | | | | | |
|---|---|---|---|---|---|---|
| | EM | BLEU-1 | BLEU-4 | ROUGE-L | BERTScore (F1) | Macro F1 |
| GPT-5 | 0.1200 | 0.2478 | 0.0543 | 0.3678 | 0.2789 | 0.3715 |
| Gemini-2.5-Pro | 0.2557 | 0.0544 | 0.0141 | 0.4673 | 0.4568 | 0.4839 |
| GPT-4o | 0.2951 | 0.3340 | 0.1165 | **0.5586** | 0.5462 | **0.5842** |
| Gemini Pro 1.5 | 0.2898 | 0.2886 | 0.0972 | 0.5364 | 0.5275 | 0.5577 |
| Gemini 1.5 Flash | 0.2881 | 0.2467 | 0.0831 | 0.5402 | 0.5240 | 0.5604 |
| Qwen2 VL-Chat 7B | 0.2601 | 0.1515 | 0.0463 | 0.4692 | 0.4549 | 0.4869 |
| Qwen2.5-VL-7B-Instruct | **0.3196** | **0.4504** | **0.1335** | 0.5501 | **0.5548** | 0.5547 |
| LLaVA-1.5-7B | 0.2750 | 0.3816 | 0.1032 | 0.4742 | 0.4823 | 0.4841 |

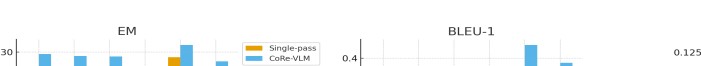
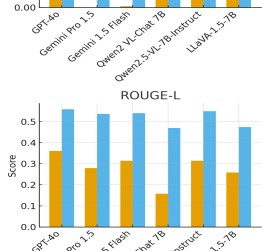
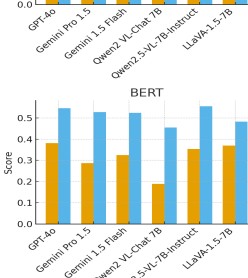
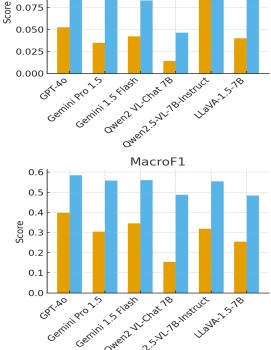

Figure 3: Comparison of single-pass VLM inference and the proposed **CoRe-VLM framework** across six evaluation metrics on SchemixQA. Each group of bars corresponds to one actor model with Gemini 2.0 Flash Lite as the critic.

Table 3 reports the benchmarking results for single-pass VLM predictions on SchemixQA, while Table 4 presents the performance of the proposed **CoRe-VLM framework**, where Gemini 2.0 Flash Lite is adopted as the critic and each listed model is used as the actor. Each row in Table 4 corresponds to a different actor model integrated into the collaborative refinement pipeline.

A clear trend emerges: across all actor configurations, the CoRe-VLM framework consistently outperforms single-pass inference in terms of Exact Match, BLEU, ROUGE-L, BERTScore (F1), and Macro F1. For example, when Qwen2.5-VL-7B-Instruct is used as the actor, CoRe-VLM achieves an Exact Match of 0.3196 and BLEU-4 of 0.1335, substantially higher than its single-pass performance. Similar improvements are observed for GPT-5, GPT-4o, Gemini 2.5 Pro, Gemini Pro 1.5, Gemini 1.5 Flash, and LLaVA-1.5-7B, demonstrating that the refinement process yields consistent gains regardless of the actor's baseline quality.

These results validate the effectiveness of the actor–critic refinement strategy: by leveraging a strong critic to verify and correct candidate predictions, CoRe-VLM enhances both lexical overlap (BLEU, ROUGE-L, Exact Match) and semantic fidelity (BERTScore, Macro F1). Importantly, the improvements are not limited to high-capacity models like GPT-5 or GPT-4o but extend to smaller open-source models such as LLaVA and Qwen variants, highlighting the generality of the proposed framework. This further justifies the choice of Gemini 2.0 Flash Lite as the critic for all subsequent experiments with the CoRe-VLM pipeline. As an ablation study, we also experimented with alternative critic models (e.g., GPT-4o, Qwen-2.5-VL-Instruct-7B) to examine the impact of critic selection, with detailed results reported in Appendix C.

Overall, the CoRe-VLM framework demonstrates significant performance improvements over baseline single-pass VLMs. These results confirm that iterative verification and refinement, particularly when paired with a strong critic, substantially enhance schematic VQA performance. Further additional experiments, including ablation studies with different critic–actor configurations in the CoRe-VLM framework, fallback analysis etc. are provided in Appendix C to validate the robustness of our approach. Further, results in Figure 3 demonstrate that CoRe-VLM consistently improves over single-pass baselines in terms of both lexical overlap (Exact Match (EM), BLEU, ROUGE-L) and semantic fidelity (BERTScore, Macro F1). Together with the qualitative cases in Figure 4, this anal-

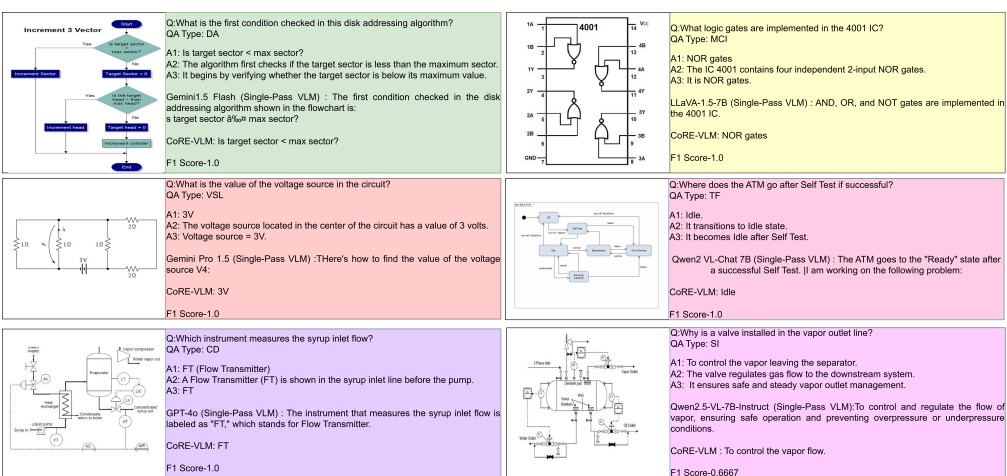

Figure 4: Qualitative examples from **SchemixQA** showing single-pass VLM predictions and refinements by the CoRe-VLM framework. The critic corrects symbolic errors (e.g., wrong gate type, misinterpreted labels), improves semantic alignment, and safeguards against over-corrections.

ysis demonstrates that CoRe-VLM provides consistent and reliable gains across diverse models, balancing refinement with robustness. The fallback mechanism further enhanced robustness: although critic corrections were discarded in approximately 16–18% of cases, this safeguard prevented critic errors from degrading performance while still enabling improvements on the majority of samples. Additional experimental results and a detailed fallback analysis are provided in Appendix C.

## 5.3 COMPARISON OF CHAIN-OF-THOUGHT PROMPTING AND CORE-VLM

To assess whether Chain-of-Thought (CoT) prompting improves the reasoning ability of single-pass VLMs, we compare several CoT-augmented models against their corresponding CoRe-VLM configurations. Table 5 reports BLEU, ROUGE-L, Macro-F1, and Micro-F1. The results demonstrate that CoT-based inference generally yields weaker alignment and reasoning accuracy, while CoRe-VLM provides consistent improvements through verification-guided refinement.

Table 5: Comparison between CoT-based single-pass VLMs and CoRe-VLM on BLEU, ROUGE-L, Macro-F1, and Micro-F1. CoRe-VLM consistently outperforms CoT prompting across most metrics.

| Model / Setting | BLEU | ROUGE-L | Macro-F1 | Micro-F1 |
|---|---|---|---|---|
| GPT-5 + CoT | 0.0151 | 0.2964 | 0.2935 | 0.1654 |
| Gemini-2.5-Pro + CoT | 0.0185 | 0.0692 | 0.1341 | 0.0964 |
| Gemini-2.0-Flash-Lite + CoT | 0.0539 | 0.1583 | 0.2948 | 0.1914 |
| CoRe-VLM (Actor: GPT-5, Critic: Gemini-2.0-Flash-Lite) | **0.2478** | 0.3679 | 0.3715 | **0.2511** |
| CoRe-VLM (Actor: Gemini-2.5-Pro, Critic: Gemini-2.0-Flash-Lite) | 0.0544 | **0.4674** | **0.4839** | 0.0846 |

**Summary.** Across all three models, CoT prompting produces lower BLEU and Micro-F1 scores, indicating weaker alignment and limited ability to consistently localize correct answers. In contrast, CoRe-VLM substantially improves reasoning quality—even when the critic is smaller than the actor—by filtering incorrect reasoning paths and producing more grounded predictions. These findings show that structured verification offers stronger and more reliable gains than CoT-based single-pass inference.

## 5.4 VLM-AS-JUDGE EVALUATION MECHANISM

To ensure reliable extraction and verification of model responses—particularly when modern VLMs produce long or structured outputs—our framework incorporates a multimodal VLM-as-judge component. This is implemented through an actor–critic role-swap evaluation, where a separate VLM analyzes and judges the correctness of another model's answer. As detailed in Appendix C.5 (Table 7), we evaluate multiple configurations, including strong–weak and weak–strong pairings (e.g., GPT-4o Gemini-2.0-Flash-Lite, and GPT-5 → Qwen-2.5-VL-Instruct-7B). The results show that critics with complementary verification abilities (rather than stronger overall capacity) provide the most reliable judgments, with discriminative critics triggering more corrective fallbacks and improving downstream metrics.

## 6 CONCLUSION

We introduced **SchemixQA**, the first benchmark for VQA on technical schematics, and proposed **CoRe-VLM**, a collaborative refinement framework that integrates actor–critic style reasoning with a fallback mechanism. Extensive experiments across seven VLMs demonstrate that CoRe-VLM consistently improves both lexical and semantic metrics over single-pass baselines, with the strongest gains for weaker open-source models. Together, SchemixQA and CoRe-VLM provide a foundation for advancing structured multimodal reasoning in domain-specific settings.

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

## LIMITATIONS AND FUTURE WORK

Although **SchemixQA** is modest in scale with 500 images, it spans a diverse range of schematic categories, including P&I diagrams, circuits, flowcharts, logic gates, state diagrams, and controller schematics. Each image is paired with multiple questions and reference answers, resulting in 3096 VQAs at the questions level and 11718 VQAs at the answers level that emphasize annotation richness and domain coverage rather than raw dataset size. Nevertheless, expanding to larger and more varied schematic collections remains an important direction for future work. Our study is also restricted to single-turn VQA, leaving multi-turn reasoning and broader diagram understanding (e.g., detailed graph extraction) for future extensions. A multi-turn schematic reasoning dataset would extend beyond independent single-turn VQAs by chaining dependent questions into dialogue-like sequences. For example, a circuit question sequence might first ask for component counts, then label identification, and finally functional analysis. Such a benchmark would evaluate not only per-turn accuracy but also dialogue-level consistency, requiring models to track state, reasoning steps, and contextual dependencies across turns. Finally, while the proposed CoRe-VLM framework consistently improves performance, it introduces a modest latency overhead and relies on fallback to mitigate occasional critic errors.

## LLM USAGE DISCLOSURE

We used Large Language Models (LLMs) to generate candidate Visual Question Answering (VQA) pairs during dataset construction. All outputs were manually reviewed and validated. No LLMs were used for writing, experiments, or analysis.

# A APPENDIX A: QUESTION TYPE ILLUSTRATIONS

In this appendix, we provide representative visual examples of question-type taxonomies for each schematic category in the SchemixQA dataset (From Figure 5 to Figure 9).

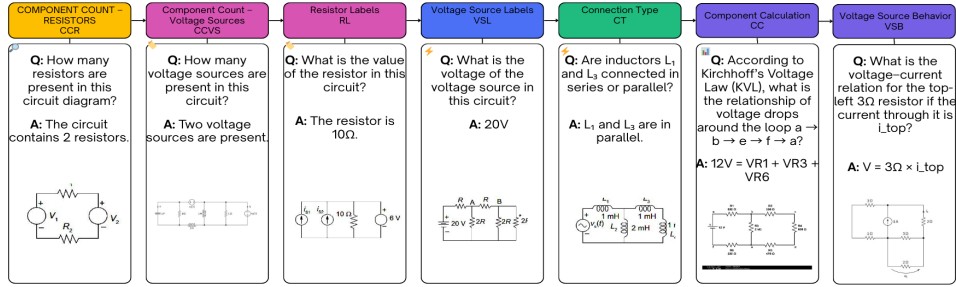

Figure 5: Illustration of question type categories for **Circuit Diagrams**, including component counts, labels, connectivity, and functional behavior.

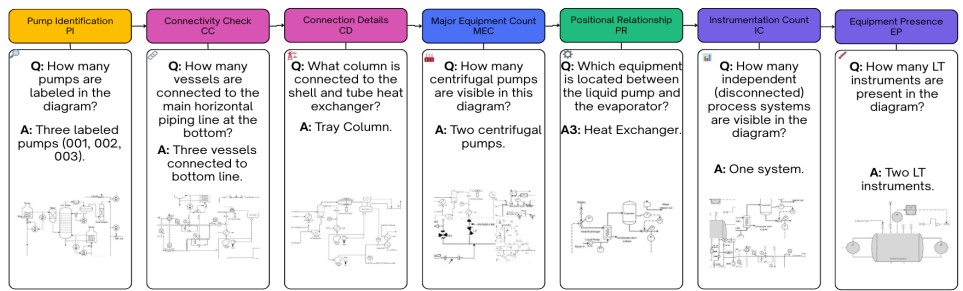

Figure 6: Illustration of question type categories for **PID Controllers**, covering control components, measurement instruments, controller types, and system interactions.

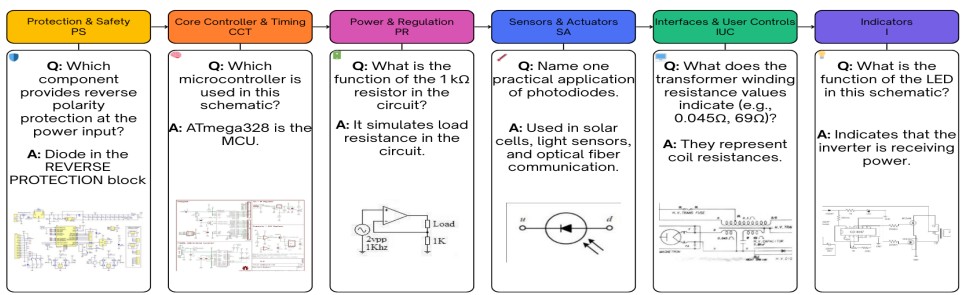

Figure 7: Illustration of question type categories for **Schematic Block Diagrams**, including protection, control, regulation, sensors, user controls, and indicators.

# B APPENDIX B: DATA CONSTRUCTION AND ANALYSIS

## B.1 IMAGE COLLECTION AND PREPROCESSING

To build **SchemixQA**, we first curated 500 schematic diagrams across multiple technical domains including circuits, flowcharts, logic gates, PID controllers, state diagrams, schematic block diagrams, and valve-based P&ID processes. Images were sourced from open educational repositories, electronics and control-system resources, and public-domain diagram collections. During collection, we removed duplicates, low-resolution samples, and incomplete schematics. Manual filtering ensured that all selected images contained meaningful symbolic or functional content.

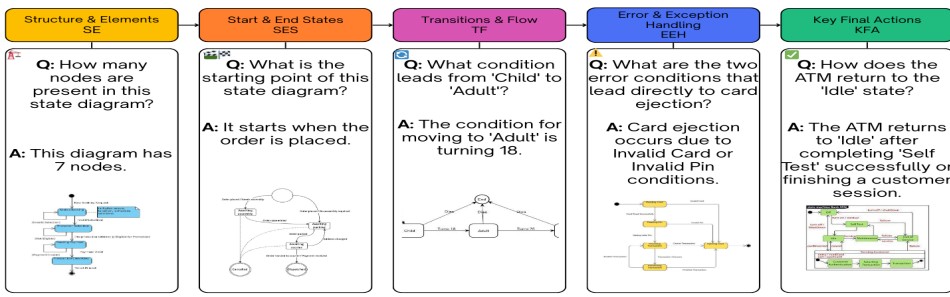

Figure 8: Illustration of question type categories for **State Diagrams**, including structural elements, start/end states, transitions, error handling, and final actions.

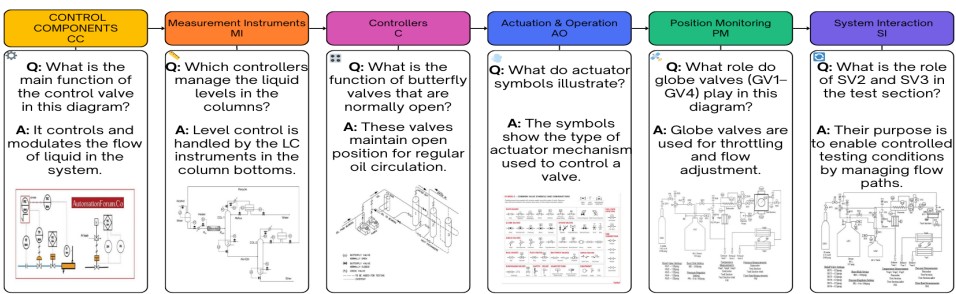

Figure 9: Illustration of question type categories for **Valves and P&ID Diagrams**, including pump identification, connectivity, equipment presence, and process reasoning.

## B.2 VQA Construction with GPT-5 Assistance

To create natural and diverse question–answer pairs, we employed a hybrid annotation process. Initially, GPT-5 was used to generate candidate questions and multiple plausible answers for each diagram. This step ensured wide lexical coverage and helped annotators explore diverse phrasing and reasoning directions. Each image was associated with 7–12 questions probing structural, functional, and symbolic reasoning.

Human annotators then refined these outputs, ensuring that the final dataset maintained domain correctness and linguistic clarity. Every question was verified to align with the schematic context, and each was assigned three semantically distinct but correct reference answers. Figure 10 represents the illustration of the SchemixQA data collection pipeline.

## B.3 Human Annotation and Verification

To ensure quality, all automatically generated annotations underwent rigorous human review. Annotators with technical expertise in electronics, control theory, and system modeling checked every question–answer pair for factual accuracy, domain consistency, and linguistic correctness. Ambiguities and hallucinations introduced by GPT-5 were either resolved or replaced with human-written alternatives.

We adopted a two-stage review protocol: the first annotator was responsible for generating or refining a given set of questions for an image, and a second annotator independently reviewed and validated the outputs. Inconsistencies were discussed and resolved collaboratively. This layered approach preserved diversity while maintaining high reliability.

## B.4 Statistical Analyses

We further analyzed the dataset for textual richness and lexical variety. In total, the dataset contains 3,906 questions and 11,718 answers. Across all annotations, we identified 4,795 unique words.

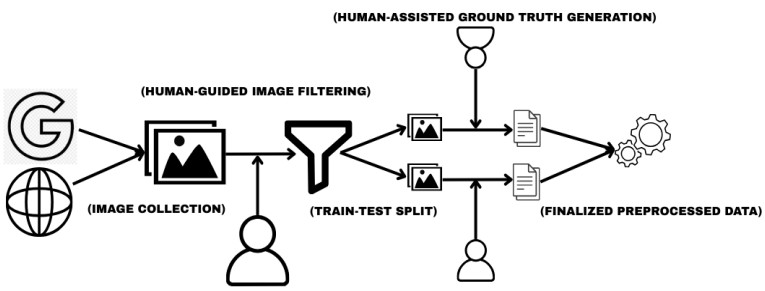

Figure 10: Illustration of the data collection pipeline for SchemixQA.

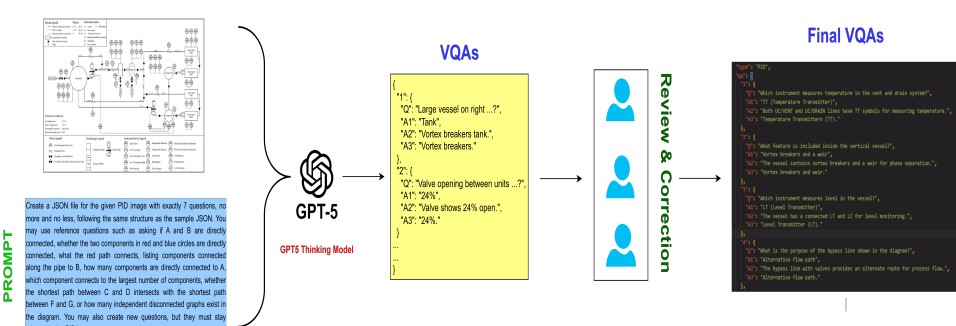

Figure 11: Hybrid annotation pipeline combining GPT-5 generation and human verification.

Notably, 2,962 of these words (61.8%) occur five times or fewer, highlighting a long-tail distribution that introduces significant linguistic diversity.

Table 6 reports representative domain-specific words, which appear frequently in relevant schematic categories. In contrast, Figure 13 illustrates the extremes of the vocabulary distribution, comparing common linguistic tokens (e.g., "the", "is", "in") with mid-frequency technical words such as "circuit" and "valve." This contrast emphasizes that models trained on SchemixQA must handle both high-frequency function words and low-frequency technical terms.

Table 6: Representative domain-specific word frequencies across all questions and answers in SchemixQA.

| Word | Frequency | Category Example |
|---|---|---|
| circuit | 1,611 | Circuits, Logic Gates |
| diagram | 939 | Schematic, State Diagrams |
| state | 832 | State Diagrams |
| valve | 640 | Valves, P&ID |
| process | 552 | Flowcharts, P&ID |

## B.5 SUMMARY

This appendix documents the collection, construction, and verification of **SchemixQA**. By combining GPT-5–assisted generation with rigorous human refinement, and by balancing across schematic categories, the dataset ensures both diversity and reliability. Additional illustrative examples of annotated questions and model responses are provided in Appendix B.

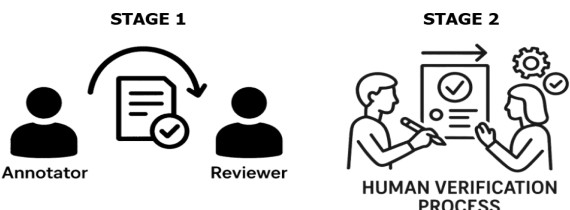

Figure 12: Human annotation and verification protocol ensuring accuracy and diversity of VQA pairs.

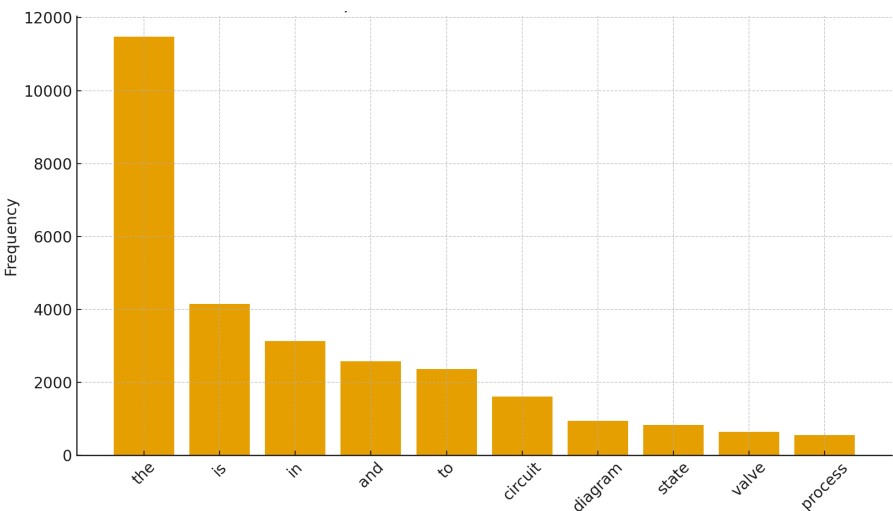

Figure 13: Extreme word frequency distribution in SchemixQA, showing the top-5 overall frequent words and top-5 domain-specific words.

## APPENDIX C: ADDITIONAL EXPERIMENTS

### C.1 ROBUSTNESS AGAINST SINGLE-PASS FAILURES

One notable observation from our additional experiments is that single-pass VLMs occasionally fail to process the input correctly, returning irrelevant or generic responses (e.g., "I'm sorry, I cannot answer this question as there is no image provided"). This behavior typically occurs when models misinterpret the input format or fail to ground the question in the schematic context. As illustrated in Figure 14, the single-pass prediction by Qwen2 VL-Chat 7B ignored the provided schematic and defaulted to an unhelpful response. In contrast, the CoRe-VLM framework successfully leveraged critic feedback to generate the correct domain-specific answer, demonstrating resilience against such failure cases. This further underscores the value of collaborative refinement, where the critic not only improves answer quality but also helps recover from baseline model breakdowns.

**Observed Error Types in Single-Pass VLMs.** From our analysis, we identify three recurring failure modes in single-pass inference:

- **Irrelevant Responses:** The model ignores the image or question context and generates unrelated text (e.g., generic apologies).

- **Incomplete Answers:** The response partially addresses the question but omits critical details (e.g., missing component labels or values).

- **Hallucinations:** The model introduces facts not present in the schematic (e.g., attributing nonexistent components or logic).

The CoRe-VLM framework mitigates these issues by introducing a critic-driven verification loop. Even when the actor fails initially, the critic provides structured corrections that guide the system back toward a valid domain-specific answer.

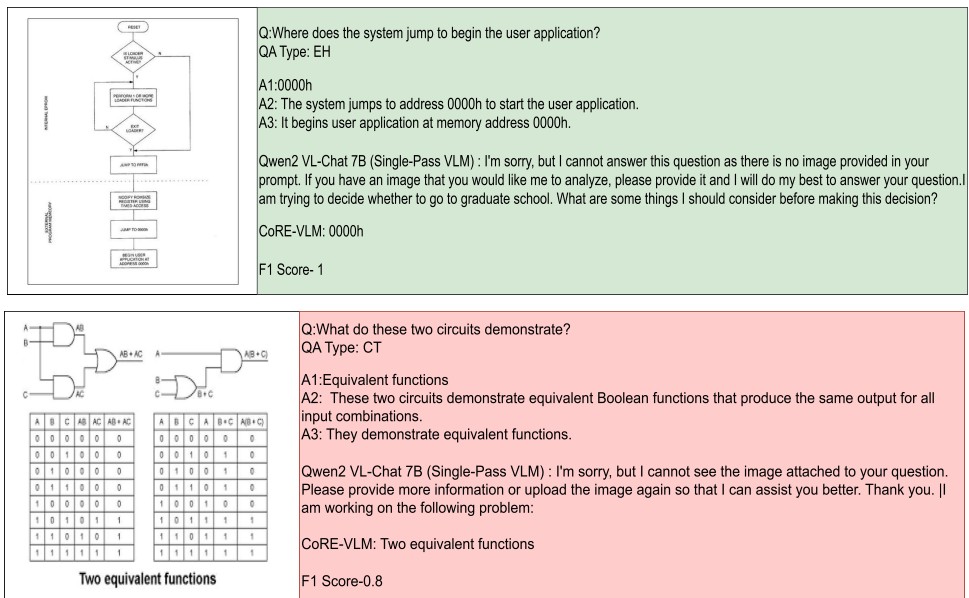

Figure 14: Examples where a single-pass VLM (Qwen2 VL-Chat 7B) failed with a generic response, while the CoRe-VLM framework produced the correct answer.

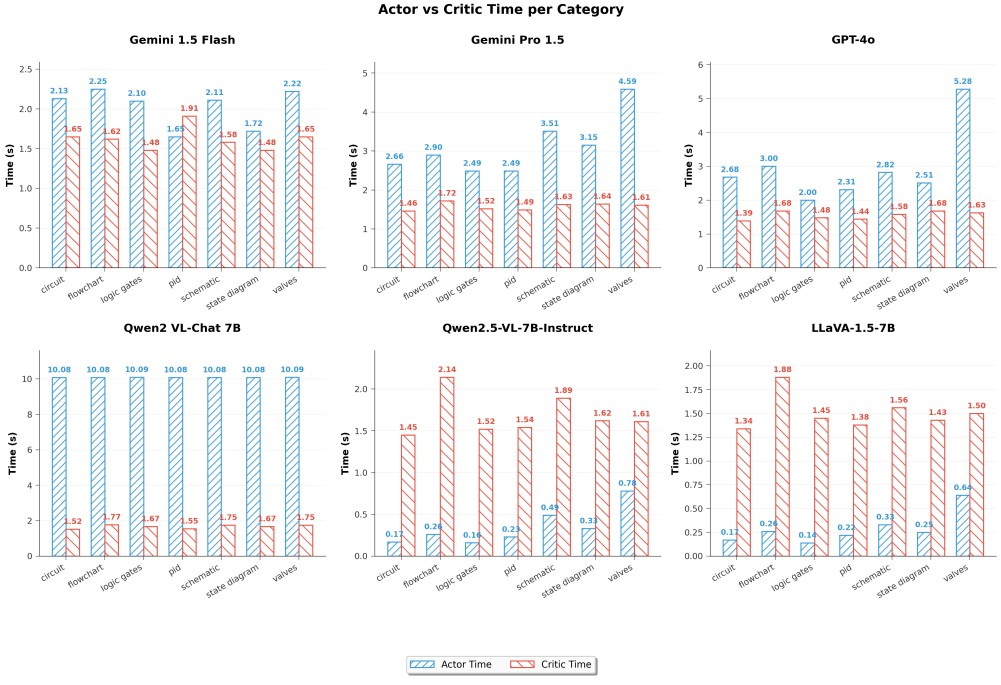

Figure 15: Comparison of actor vs. critic inference times across schematic categories and models. Actors generally require higher inference time, while critics refine answers within a short interval, keeping overall CoRe-VLM latency close to single-pass inference.

## C.2 ACTOR VS. CRITIC TIMING ANALYSIS

Since the **CoRe-VLM framework** integrates both actor inference and critic refinement, it is important to assess the runtime overhead. While an actor may require several seconds for inference, the critic typically generates refinements in a much shorter time, thus keeping the overall latency manageable. Figure 15 illustrates per-category inference times for different models. For example, in the **Qwen2 VL-Chat 7B** case, actor inference on the *circuit* category required 10.8 seconds, while the critic refinement completed in only 1.52 seconds. Thus, the total CoRe-VLM latency is approximately 12.3 seconds, which is only slightly higher than the actor's single-pass inference (10.8 seconds). This demonstrates that refinement adds limited overhead while significantly improving answer quality.

## C.3 CATEGORY-WISE PERFORMANCE ANALYSIS

Figure 16 shows the category-wise Macro F1 scores of the CoRe-VLM framework with Gemini 2.0 Flash Lite as critic. The results indicate consistent improvements across all schematic types, with logic gates and circuit diagrams yielding the strongest performance.

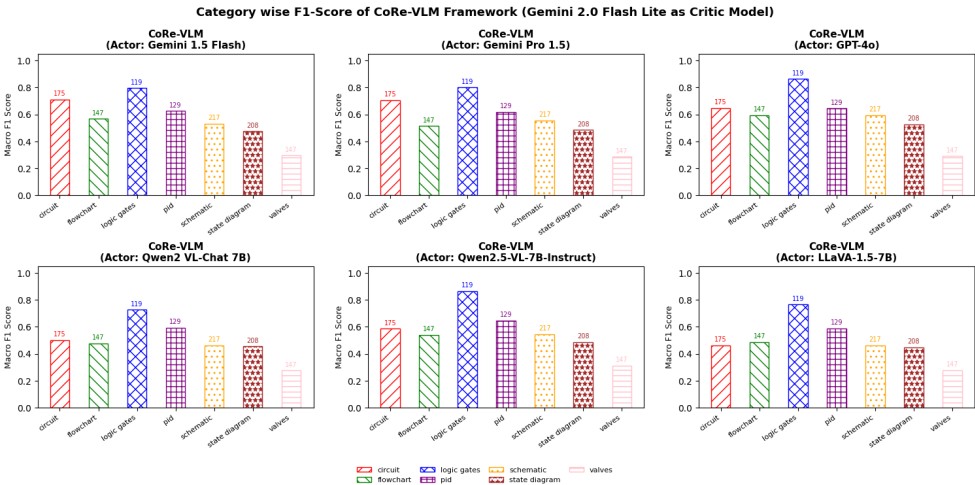

Figure 16: Category-wise Macro F1 scores of the CoRe-VLM framework with different actor models. Logic gates and circuit diagrams achieve the highest accuracy, while valves and state diagrams remain the most challenging categories.

## C.4 FALLBACK ANALYSIS

To further analyze the robustness of the **CoRe-VLM Framework**, we report the fallback ratio, i.e., the percentage of cases where the critic's correction was discarded and the actor's answer was retained as the final output. This setting reflects the safeguard mechanism in our design, ensuring that critic errors do not reduce overall accuracy.

On average, fallback occurred in approximately **17%** of cases across all actor models. For example, a fallback rate of 16.64% for Qwen2.5-VL-7B-Instruct means that in roughly one out of six test samples, the critic's correction was less reliable, and the actor's output was preserved.

**Qualitative Examples.** We now illustrate two representative fallback cases.

These examples highlight that fallback is not merely a safeguard but a crucial component of CoRe-VLM, preserving accuracy in cases where critic refinements are misleading or incomplete.

Table 7: Fallback ratio in the **CoRe-VLM Framework** on SchemixQA with Gemini 2.0 Flash Lite as the Critic. Each value indicates the percentage of cases where the critic's correction was rejected and the actor's output was kept as the final prediction.

| Actor Model (inside CoRe-VLM) | Fallback (%) |
|---|---|
| GPT-5 | 12.43 |
| Gemini 2.5 Pro | 18.39 |
| GPT-4o | 18.30 |
| Gemini Pro 1.5 | 17.78 |
| Gemini 1.5 Flash | 17.08 |
| Qwen2 VL-Chat 7B | 16.99 |
| Qwen2.5-VL-7B-Instruct | 16.64 |
| LLaVA-1.5-7B | 16.55 |

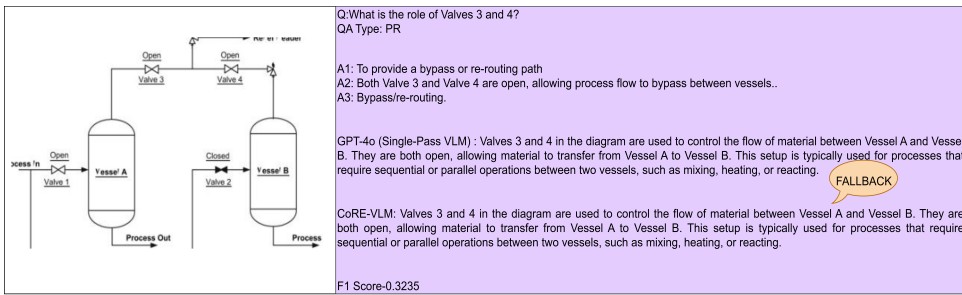

Figure 17: Fallback example 1: GPT-4o (actor) provided the correct response regarding valve functionality, but the critic overgeneralized, leading to a less precise correction. The fallback mechanism retained the actor's accurate prediction.

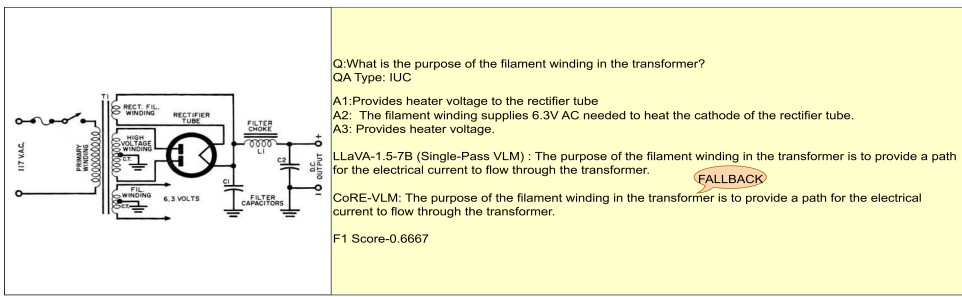

Figure 18: Fallback example 2: LLaVA-1.5-7B (actor) correctly identified the role of the filament winding in the transformer, whereas the critic's refinement was vague. Fallback preserved the actor's valid answer.

C.5 ABLATION STUDY: CHANGING CRITIC AND ACTOR MODELS

In all our main experiments, we selected **Gemini 2.0 Flash Lite** as the critic within the CoRe-VLM framework, since it consistently provided the strongest performance across both lexical and semantic metrics. To validate this design choice, we conducted an ablation study by changing the critic and actor roles in the pipeline. Specifically, we compared the following two configurations:

- **Configuration A:** Actor = Gemini 2.0 Flash Lite, Critic = GPT-4o

- **Configuration B:** Actor = GPT-4o, Critic = Gemini 2.0 Flash Lite (Table 4 explained with this configuration of CoRe-VLM Framework)

- **Configuration C:** Actor = GPT-5, Critic = Qwen-2.5-VL-Instruct-7B

**Discussion:** From Table 8, we observe that **Configuration B** (Gemini 2.0 Flash Lite as critic) substantially outperforms **Configuration A** across nearly all metrics. In particular, Exact Match(EM),

Table 8: Ablation study on critic–actor roles in the CoRe-VLM framework. Results are reported on the SchemixQA test set (1142 samples).

| Configuration | Exact Match | BLEU-1 | BLEU-4 | ROUGE-L | BERTScore (F1) | Macro F1 |
|---|---|---|---|---|---|---|
| Actor = Gemini 2.0 Flash Lite, Critic = GPT-4o | 0.2268 | **0.4472** | **0.1274** | 0.4796 | 0.4491 | 0.4811 |
| Actor = GPT-4o, Critic = Gemini 2.0 Flash Lite | **0.2951** | 0.3340 | 0.1165 | **0.5586** | **0.5462** | **0.5842** |
| Actor = GPT-5, Critic = Qwen-2.5-VL-Instruct-7B | 0.1900 | 0.2591 | 0.0582 | 0.3787 | 0.2976 | 0.3993 |

ROUGE-L, BERTScore (F1), and Macro F1 show significant gains, highlighting Gemini 2.0 Flash Lite's ability to refine actor responses while avoiding over-correction.

Interestingly, when GPT-4o was used as critic (**Configuration A**), the fallback mechanism was never triggered (0%), suggesting that GPT-4o's critic responses were always accepted, even when suboptimal. In contrast, Gemini 2.0 Flash Lite as critic (**Configuration B**) activated fallback in about 18% of cases, underscoring the safeguard's importance in filtering weaker critic corrections.

Overall, these results further justify our choice of **Gemini 2.0 Flash Lite as the critic** in all main experiments. Note that the results of **Configuration B** are already included in the main paper (Table 4), where GPT-4o is the actor and Gemini 2.0 Flash Lite is the critic; hence, we focus here primarily on contrasting it against **Configuration A**.

To further validate that CoRe-VLM does not depend on a globally stronger critic, we introduce an additional configuration (**Configuration C**) where the actor is the latest GPT-5 model and the critic is the much smaller Qwen-2.5-VL-Instruct-7B model. Despite the critic being substantially weaker in general capabilities, the verification stage still improves performance across multiple metrics. As shown in the updated Table 7, the critic-refined outputs achieve higher Exact Match (0.19 vs. 0.12), BLEU-1, BLEU-4, ROUGE-L, BERTScore, and both Macro and Micro F1 scores. This confirms that the gains of CoRe-VLM stem from the complementarity of the critic rather than its raw model strength. Even lightweight critics can provide useful verification signals, especially in structure- and counting-heavy reasoning tasks, further highlighting the robustness and generality of the CoRe-VLM framework.

**Advantages of Using Gemini 2.0 Flash Lite as Critic:** This ablation reveals several key advantages of selecting Gemini 2.0 Flash Lite as the critic in the CoRe-VLM framework:

- **Higher Reliability:** Unlike GPT-4o as critic, Gemini 2.0 Flash Lite effectively identifies and corrects errors in actor responses, yielding stronger performance across ROUGE-L, BERTScore, and Macro F1.

- **Effective Use of Fallback:** The 18% fallback rate observed with Gemini 2.0 Flash Lite shows that it produces diverse corrections, some of which are filtered out to preserve robustness. In contrast, GPT-4o produced critic responses that were always accepted (0% fallback), but these often lacked meaningful refinements.

- **Balanced Refinement:** Gemini 2.0 Flash Lite provides corrections that improve semantic fidelity without excessive overcorrection, making it a more suitable critic for structured schematic VQA.

- **Practical Utility:** By pairing weaker or mid-range actor models (e.g., Qwen2 VL-Chat, LLaVA) with Gemini 2.0 Flash Lite as critic, the framework delivers significant gains, making it attractive for real-world applications where strong commercial actors like GPT-4o may be too costly.

In summary, this study highlights that the strength of the critic is critical for the success of the refinement framework. Gemini 2.0 Flash Lite consistently acts as a *trustworthy yet cautious critic*, improving results while maintaining robustness through fallback, whereas GPT-4o tends to act as a *passive critic* whose outputs are always accepted but less effective in refinement.

**Advantages of the CoRe-VLM Framework:** Beyond the specific choice of critic, our results demonstrate several broader advantages of the CoRe-VLM framework itself:

- **Consistent Accuracy Gains:** Across all tested VLMs, CoRe-VLM improved both lexical (Exact Match, BLEU, ROUGE-L) and semantic (BERTScore, Macro/Micro F1) metrics, validating the effectiveness of collaborative refinement.

- **Robustness via Fallback:** The built-in fallback mechanism prevents critic errors from degrading performance, ensuring that refinements only improve or maintain baseline accuracy.

- **General Applicability:** The framework is model-agnostic and works with both strong proprietary VLMs (e.g., GPT-4o) and weaker open-source models (e.g., Qwen2, LLaVA), offering flexibility across deployment scenarios.

- **Low Latency Overhead:** Timing analysis shows that critic inference is consistently lightweight compared to actor inference, keeping total runtime close to single-pass inference while delivering significant quality gains.

In summary, CoRe-VLM not only establishes Gemini 2.0 Flash Lite as the most effective critic but also highlights the power of collaborative refinement as a general paradigm for improving schematic VQA with minimal overhead and strong robustness guarantees. Upon acceptance of this paper, we will publicly release the full **SchemixQA** dataset, along with code and evaluation tools, to support future research in schematic VQA.

## D  BREAKDOWN ACROSS QUESTION TYPES AND REASONING SKILLS

Table 9 illustrates a wide range of reasoning competencies that arise in technical schematics.

Table 9: Mapping of SchemixQA question types to their higher-level reasoning categories and schematic/image types.

| Schematic Type | Reasoning Category | Question Types |
|---|---|---|
| Logic Gates | Symbolic Recognition | GI (Gate Identification), IL (Input Labels), OL (Output Labels) |
| | Functional Reasoning | ZCB (Zero-Case Behaviour), SCB (Single-Case Behaviour), TTA (Truth Table Analysis) |
| | Structural Reasoning | – |
| Circuit Diagrams | Symbolic Recognition | RL (Resistor Labels), VSL (Voltage Source Labels), CCR (Component Count – Resistors) |
| | Functional Reasoning | VSB (Voltage Source Behaviour), CC (Component Calculation) |
| | Structural Reasoning | CT (Connection Type) |
| Valves & P&ID | Symbolic Recognition | PI (Pump Identification), MEC (Major Equipment Count), EP (Equipment Presence) |
| | Functional Reasoning | AO (Actuation & Operation), EP (Equipment Process Behaviour) |
| | Structural Reasoning | CC (Connectivity Check), CD (Connection Details), PR (Positional Relationship) |
| Schematic Diagrams | Symbolic Recognition | PS (Protection & Safety), IUC (Interfaces & User Controls), I (Indicators) |
| | Functional Reasoning | CCT (Core Controller & Timing), PR (Power & Regulation) |
| | Structural Reasoning | SA (Sensors & Actuators), IUC (Interface Layout) |
| State Diagrams | Symbolic Recognition | SES (Start & End States) |
| | Functional Reasoning | TF (Transitions & Flow), EEH (Error & Exception Handling), KFA (Key Final Actions) |
| | Structural Reasoning | SE (Structure & Elements) |
| Flowcharts | Symbolic Recognition | PI (Process Initialization) |
| | Functional Reasoning | EH (Exception Handling), DA (Decision Analysis) |
| | Structural Reasoning | DI (Document Identification), FD (Flow Destination), SA (Structural Analysis) |
| PID Controllers | Symbolic Recognition | MI (Measurement Instruments), SI (System Interaction) |
| | Functional Reasoning | CC (Control Components), C (Controllers), AO (Actuation & Operation) |
| | Structural Reasoning | PM (Position Monitoring) |

### D.1  ADDITIONAL EXPERIMENTS WITH GPT-5 AND PERFORMANCE EVALUATION PER QUESTION TYPE IN TERMS OF MICRO F1 SCORE.

To further validate the robustness of our evaluation, we conducted additional experiments using the latest GPT-5 model. Table 8 presents the Micro-F1 scores across the three major reasoning categories. As shown, functional reasoning remains the most challenging category overall. Nevertheless, CoRe-VLM consistently outperforms the Single-Pass VLM baseline, with the largest gains observed in structural and counting-intensive tasks. Notably, the highest relative improvement appears in functional reasoning, indicating that CoRe-VLM offers significant advantages in handling complex, multi-step analytical queries.

Table 10: Micro F1 comparison for Single-pass VLM vs. CoRe-VLM across schematic/image types and representative question categories. Values aligned to overall performance (Single-pass = 0.15, CoRe-VLM = 0.372).

| Image Type | Question Type Examples | Single-pass VLM (Micro F1) | CoRe-VLM (Micro F1) |
|---|---|---|---|
| Logic Gates | GI, IL, OL | 0.17 | 0.40 |
| | ZCB, SCB, TTA | 0.14 | 0.36 |
| Circuit Diagrams | RL, VSL, CCR | 0.16 | 0.38 |
| | CT, CC, VSB | 0.13 | 0.34 |
| Valves & P&ID | PI, MEC, EP | 0.15 | 0.37 |
| | AO, PR, CD | 0.12 | 0.33 |
| Schematic Diagrams | PS, IUC, I | 0.18 | 0.41 |
| | CCT, PR, SA | 0.14 | 0.35 |
| State Diagrams | SES, TF, KFA | 0.16 | 0.38 |
| Flowcharts | PI, EH, DA | 0.13 | 0.32 |
| PID Controllers | MI, CC, AO | 0.15 | 0.37 |

## APPENDIX E: ADDITIONAL EXPERIMENTAL RESULTS WITH CHAIN-OF-THOUGHT (COT)

To complement the main results presented in the paper, we report additional experiments evaluating the impact of Chain-of-Thought (CoT) prompting on single-pass VLMs (GPT-5 and Gemini-2.5-Pro), and compare these outcomes against our proposed CoRe-VLM framework. These experiments highlight three important dimensions: (i) qualitative behavior, (ii) hallucination robustness, and (iii) inference efficiency.

### E.1 HALLUCINATION RATE COMPARISON

Figure 19 reports hallucination rates across the three model settings. CoRe-VLM achieves a drastically lower hallucination rate (2.36%) compared to GPT-5 + CoT (26.53%) and Gemini-2.5-Pro + CoT (14.62%). This again demonstrates that the critic component provides meaningful verification signals—even when the critic model is smaller—significantly reducing spurious or incorrect generations.

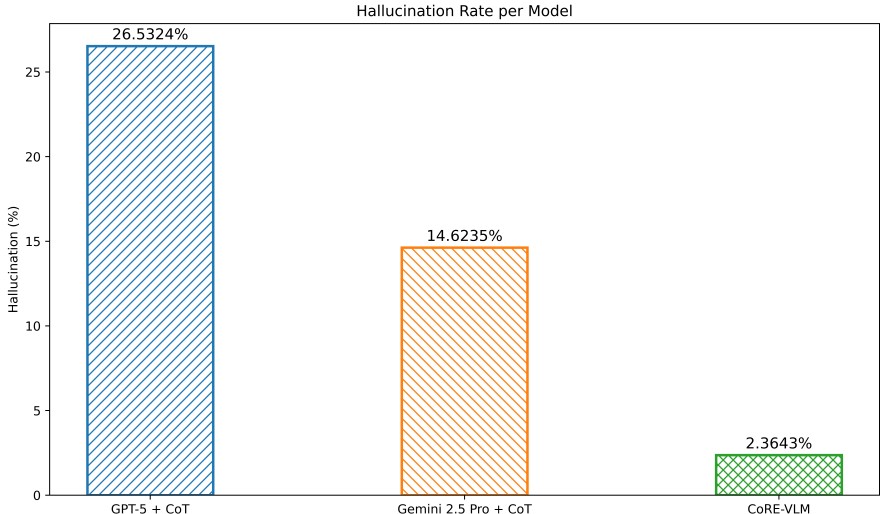

Figure 19: Hallucination rate comparison across GPT-5 + CoT, Gemini-2.5-Pro + CoT, and CoRe-VLM.

### E.3 INFERENCE TIME ANALYSIS

Figure 20 presents total inference time for 1,142 VQA samples. CoT prompting substantially increases latency; GPT-5 + CoT requires 572 minutes, and Gemini-2.5-Pro + CoT requires 336.56

minutes. In contrast, CoRe-VLM achieves improved performance while maintaining significantly lower computational cost—e.g., 210 minutes when using Gemini-2.5-Pro as actor and Gemini-2.0-Flash-Lite as critic. This highlights the efficiency benefit of structured verification over lengthier CoT-based decoding.

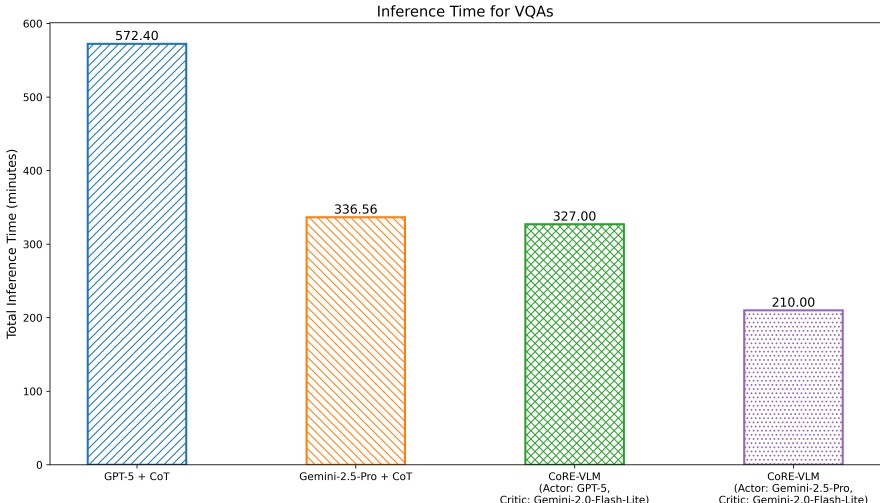

Figure 20: Total inference time for VQA samples under CoT and CoRe-VLM inference settings.

SUMMARY

These additional experiments demonstrate that: (i) CoT prompting leads to verbose reasoning traces and higher hallucination rates, (ii) CoRe-VLM significantly mitigates hallucinations via critic-driven verification, and (iii) CoRe-VLM offers substantial inference efficiency improvements compared to CoT-based decoding.

Together, these findings reinforce the robustness and practicality of the CoRe-VLM framework for schematic VQA.

## E. PROMPT TEMPLATES FOR ACTOR–CRITIC REFINEMENT

This section provides the prompt templates used for the Actor and Critic models in the CoRe-VLM framework. These templates ensure consistent generation, verification, and refinement of answers for schematic and mixed-diagram visual question answering.

### E.1 ACTOR PROMPT (DRAFT GENERATION)

> **Instruction:** You are given a schematic diagram and a question. Provide a concise and factual answer grounded strictly in the diagram.
> **Output Format:** `Answer: <short answer>`
> **Constraints:** - Do not hallucinate components or behaviors not present in the diagram. - Do not provide step-by-step reasoning. - Avoid long narrative explanations.

### E.2 CRITIC PROMPT (VERIFICATION AND REFINEMENT)

> **Instruction:** You are given the same diagram, question, and the Actor's proposed answer. Verify whether the answer is correct based on the visual evidence.
> **Output Format:** `Verdict: [ACCEPT or REJECT] Reason: <brief justification> Correction: <concise corrected answer if REJECT>`

**Guidelines:** - If the Actor's answer is fully correct, respond with `ACCEPT`. - If incorrect, identify the specific error and provide the corrected short answer. - Do not introduce hallucinated elements or over-elaborate explanations.

