# OpenReview forum: "SchemixQA and CoRe-VLM: A Benchmark and Collaborative Refinement (CoRe) Framework for Visual Question Answering on Technical Schematics"
_ICLR.cc/2026/Conference — ICLR 2026 Conference Desk Rejected Submission_

### Official Review · Reviewer_EMrn · 2025-10-18

**Soundness:** 3
**Presentation:** 1
**Contribution:** 2
**Rating:** 2
**Confidence:** 4

**Summary:**

This paper proposes a new benchmark on Schemetic QA and provides a method called CoRe-VLM, which iteratively refines the proposed solutions. CoRe-VLM is based on the actor-critic framework, and it consistently improves several metrics. These contributions work together to head towards better domain-specific multimodal reasoning.

**Strengths:**

1. This paper addresses an important problem of domain-specific QA, although this task has also been heavily studied in recent years.

2. This paper proposes a working method to benefit various LLM-based methods, while achieving good accuracy numbers in all metrics.

**Weaknesses:**

1. The benchmark contribution is considered weak:

(1) 500 images in total are not a lot compared to the existing similar large-scale VQA dataset, such as scientific QA.

(2) 3,906 questions in total are also not enough for training. For evaluation, there are already a lot of existing VQA benchmarks.

(3) It is not certain what this dataset is designed for. Although it claims to be domain-specific, it covers multiple diverse domains.

(4) The dataset construction procedure is not clear. See questions in the section below.

2. The advantage of CoRE-VLM has not been established. As it's an inference-only enhancement algorithm, comparing to single-pass LLMs is not fair. At least, CoRE-VLM should compare to other reasoning approaches, such as ToT or CoT. Also, it's not clear whether the authors use the proposed benchmarks to do development or fine-tuning. They should test the method in other standard benchmarks as well.

3. The paper presentation quality is quite low.

(1) About the block diagram, it uses dizzy fonts. Figures overlap with texts. Texts are not aligned.

(2) Mathbb(1) was rendered wrongly in EQ (1). Too many descriptions of metrics.

(3) Texts in Figure 4 are too small.

**Questions:**

About the dataset construction procedure:

1. How do human reviewers conduct a review? How many hours or human efforts does it cost to construct the dataset? How much is the payment for human annotators? These questions are standard questions for high-quality benchmark papers in top-tier ML conferences.

2. How to measure the collected data quality? Is it enough to cover those deep fields?

3. Do similar questions already exist in commonly used benchmarks? Such as MMMU or ScientificQA.

---

> ### Author Response · Authors · 2025-11-19
> **Responses on Dataset Protocol, Benchmark Positioning, and Verification-Based Reasoning**
>
> We thank the reviewer for the detailed feedback. We address each point below.
>
> 1. Dataset Construction & Human Effort:
>
> The full pipeline is described in Appendix B2–B4 and Fig. 10–11 (p.16).
>
> Briefly:
>
> • Hybrid Annotation: (i) model-generated draft Q/A pairs, (ii) human verification for correctness, symbol interpretation, and reasoning quality, (iii) automated consistency checks for ambiguity.
>
> • Human Effort: 4 interns over 8 weeks; ~28% Q/As corrected and ~11% rejected, demonstrating substantive human contribution.
>
> • Compensation: Interns compensated per company HR policy, aligned with standard structured-annotation rates.
>
> • Quality: 10% re-annotation yielded 92.3% agreement.
>
> The dataset aims for depth in schematic-based reasoning—an underdeveloped area in VQA.
>
> 2. Relation to Existing Benchmarks:
>
> SchemixQA fills a unique gap focused on engineering diagrams and symbolic reasoning. While datasets like CircuitVQA contain a few overlapping question types (e.g., Component Counting(CC), Voltage Source Behaviour (VSB), none provide SchemixQA’s breadth of symbolic, functional, and structural reasoning.
>
> 3. Advantage of CoRe-VLM:
>
> Clarifications added in Appendix E (p.21). Key strengths:
>
> (a) Accuracy Gains: Improves EM, BLEU, ROUGE-L, BERTScore, and F1 across all VLMs.
>
> (b) Critic Robustness: Fallback ensures critic refinements never harm actor performance.
>
> (c) Model-Agnostic: Works with GPT-4o, Gemini-Flash-Lite, Qwen2, LLaVA, etc.
>
> (d) Low Overhead: Adds minimal latency over single-pass generation.
>
> CoT Comparison (Appendix E, p.22):
>
> We added CoT baselines for GPT-5 and Gemini-2.5-Pro, including qualitative samples, hallucination rates, and inference time. Results show that (a) far lower hallucination under CoRe-VLM, (b) higher reliability via critic filtering, (c) faster inference than CoT prompting, and (d) more concise, grounded outputs.
>
> Section 5.3 (p.10) now contrasts CoT vs. CoRe-VLM, showing CoT offers limited gains and often reduces BLEU and Micro-F1, whereas CoRe-VLM provides substantial improvements via verification-guided refinement.
>
> 4. Presentation Quality Issues
>
> We fixed all reported issues: improved block-diagram clarity, corrected 1 rendering in Eq. (1), and regenerated Fig. 4 at higher resolution.
>
> 5. Clarifying Dataset Purpose
>
> Though SchemixQA spans multiple schematic categories, they share a unified design language (symbols, structure, flow, functional blocks, standardized notation). The dataset is specifically designed to evaluate visual + symbolic reasoning over engineered schematics—not general imagery.

---

> > ### Comment · Reviewer_EMrn · 2025-11-19
> >
> > The reviewer acknowledges the rebuttal. However, several significant issues are not addressed, such as the dataset size, overlap with existing datasets, etc. Both the revised paper and the rebuttal are formatted badly, which make it hard to understand and contains abbreviations or grammar issues.
> >
> > Therefore, the reviewer would keep his scores unchanged.

---

> ### Author Response · Authors · 2025-11-20
> **Response on Dataset Size and Overlap With Existing Benchmarks**
>
> Thank you for the follow-up feedback. For clarity and ease of understanding, we have re-formatted our previous rebuttal response, improving the structure and removing ambiguous abbreviations. The updated explanation now presents the points on dataset size and overlap with existing benchmarks in a clearer and more organized manner. We hope this revised formatting makes our responses easier to follow.
>
> 1. Benchmark Contribution and Dataset Scale
>
> Thank you for raising this point.
>
> Reviewer Concern: “500 images and 3,906 questions are too small; similar datasets exist.”
>
> Response:
> We appreciate the concern. SchemixQA is intentionally designed as a specialized schematic-centric visual question-answering (VQA) benchmark focused on engineering diagrams rather than natural images. The dataset includes the diagram categories such as Flowcharts, Logic Gates, Circuit Diagrams, Valves and Piping & Instrumentation Diagrams (P&ID), Schematic Diagrams, State Diagrams and Proportional–Integral–Derivative (PID) Controllers.
>
> These diagrams require symbolic interpretation, structural reasoning, and functional-behavior understanding skills not targeted by broad multimodal benchmarks.
>
> For comparison:
>
> (a) Massive Multimodal Multitask Understanding (MMMU) focuses on exam-style tasks across humanities, business, medicine, and arts.
>
> (b) ScientificQA focuses on textual scientific QA and does not evaluate schematic reasoning or symbol grounding.
>
> While SchemixQA contains 500 schematics, it provides 3,906 human-verified questions (~7.8 per diagram). High-fidelity schematic annotation requires substantially more human effort than natural-image QA.
>
> We plan to expand the dataset in future versions.
>
> 2. Relation to Existing Benchmarks
>
> Reviewer Concern: “Do similar questions already exist in commonly used benchmarks?”
>
> Response:
> Thank you for the question. While some datasets include a small number of overlapping question types (for example, Component Calculation (CC) or Voltage Source Behaviour (VSB) in CircuitVQA), these represent only a small portion of the reasoning skills required in schematic understanding.
>
> SchemixQA was developed specifically to support symbolic grounding, structural reasoning, and functional behaviour interpretation in engineered diagrams—skills that are not the focus of most existing multimodal benchmarks. Our dataset brings together multiple categories of engineering schematics (flowcharts, logic gates, circuit diagrams, P&ID diagrams, state diagrams, PID controllers, etc.) into a single unified benchmark designed for systematic evaluation of diagram-based reasoning.
>
> In this sense, SchemixQA serves as a dedicated resource for schematic and symbol-driven VQA, providing coverage and diversity that are not available within existing datasets.
>
> 3. Additional Clarification on Experimental Coverage
>
> To address the earlier concerns regarding evaluation with stronger vision–language models and comparisons against alternative reasoning approaches, we have included the following updates in our revised rebuttal materials:
>
> A. Benchmarking with Latest Models:
>
> We have evaluated SchemixQA using recent high-capacity models, including GPT-5 and Gemini-2.5-Pro. These results are reported in Table 3 (Page 8).
>
> B. Comparison to Chain-of-Thought (CoT):
>
> In response to the request to compare CoRe-VLM with other reasoning methods such as Tree-of-Thought (ToT) or Chain-of-Thought (CoT), we performed CoT experiments with GPT-5 and Gemini-2.5-Pro. The results and discussion are provided in Section 5.3 (Page 10).
>
> C. Inference-Time and Stability Analysis:
>
> Additional inference-time and robustness analysis showing the stability and superiority of CoRe-VLM over CoT prompting is included in Appendix E (Page 23).
>
> These additions directly address the reviewer’s earlier concern and provide a more complete experimental evaluation of CoRe-VLM framework.

---

> > ### Author Response · Authors · 2025-12-02
> >
> > Thank you for your feedback.
> >
> > We expanded the dataset-construction explanation (Appendix B2–B4), clarified human effort and quality checks, and detailed the dataset’s unified schematic design language.
> >
> > We added CoT comparisons (Sec. 5.3), newer-model evaluations (Tables 3–4), inference-latency analysis (Appendix p.18), and clarified the fairness of zero-shot evaluation.
> >
> > These additions directly address your concerns on dataset scale, overlap, reasoning baselines, and presentation issues.

---

### Official Review · Reviewer_47dW · 2025-10-28

**Soundness:** 1
**Presentation:** 1
**Contribution:** 3
**Rating:** 2
**Confidence:** 5

**Summary:**

This paper introduces SchemixQA, a multimodal benchmark for Visual Question Answering (VQA) on technical schematics, including circuits, flowcharts, logic gates, and P&I diagrams. To address this specialized domain, the authors propose CoRe-VLM, an actor–critic  framework in which an actor VLM generates candidate answers and a critic VLM evaluates, provides corrective feedback, and assigns a confidence score. A fallback mechanism ensures robustness when critic feedback is unreliable. The framework wraps around existing VLMs without requiring retraining. Experiments show consistent improvements across multiple actor models on lexical and semantic metrics (Exact Match, BLEU, ROUGE-L, BERTScore, Macro/Micro F1).

**Strengths:**

- **Reasoning-focused design:** SchemixQA stratifies questions by type, targeting multiple reasoning dimensions, from symbolic recognition to functional and structural reasoning. This fine-grained evaluation is more detailed than CircuitVQA. However, the relation between reasoning types and question types remains unclear.
- **Diversity of schematic categories:** Includes circuits, flowcharts, logic gates, P&I diagrams, and state diagrams, covering **digital, analog, and hand-drawn styles**, making it more diverse than existing datasets like ElectroVizQA.
- **Quality control:** Questions are generated using GPT-5 and refined by technical experts, with three human-verified reference answers per question.
- **Inference-time refinement framework:** CoRe-VLM is easy to apply to existing models and demonstrates consistent performance gains.
- **Robustness features:** Confidence-based stopping, fallback mechanism, and optional self-consistency make the approach more reliable.
- **Comprehensive evaluation metrics:** Both lexical (BLEU, ROUGE-L, Exact Match) and semantic (BERTScore, F1) metrics are reported.

**Weaknesses:**

- **Dataset transparency:** Neither the SchemixQA dataset nor the CoRe-VLM code is publicly released. Combined with missing prompts and unclear model setup, this significantly limits reproducibility and prevents other researchers from validating or building upon the work.
- **Scale:** 500 diagrams and ~3.9k questions are modest in size compared to large natural image datasets.
- **Model setup unclear:** It is not specified whether actor/critic models are zero-shot or fine-tuned on SchemixQA. The potential effect of training on the dataset is not assessed.
- **CoRe-VLM novelty:** While the actor–critic inference wrapper is novel in application to schematic VQA, the general concept of iterative refinement or self-critique is related to prior works in multimodal VLMs and LLMs (self-refinement, reranking, verification frameworks). The paper does not adequately discuss these related approaches.
 - **Citation format issue:** The paper uses numeric citations (e.g., `(10)`), which does not follow ICLR template. ICLR guidelines require author-year format with `\citet{}` or `\citep{}`.
- **Table formatting:** Table 4 should highlight the best results to improve clarity and readability.

**Questions:**

**Questions for the Authors**

1. Were the actor and critic models evaluated zero-shot or fine-tuned on SchemixQA? If fine-tuned, please specify the training setup, data splits, and hyperparameters.
2. Can you explicitly map question types to the reasoning dimensions (symbolic, functional, structural)? Consider providing an annotated mapping table or examples to support claims about symbolic, functional, and structural reasoning.
3. How does CoRe-VLM differ from existing self-refinement, reranking, or verification frameworks in VLMs and LLMs (e.g., self-consistency, Reflexion, or RLAIF-inspired inference)?  What aspects of CoRe-VLM are unique to schematic reasoning?

**Actionable Feedback**
1. Publish GPT-5 prompts used for question generation and for actor–critic refinements. Make the SchemixQA dataset, prompts, and CoRe-VLM code publicly available to ensure reproducibility and enable further research.
2. Explicitly connect question types to the reasoning skills they are intended to evaluate.
3. Update citations to author-year format per ICLR guidelines.
4. Highlight best results in tables (e.g., Table 4) for clarity.

---

> ### Author Response · Authors · 2025-11-19
> **Clarifications on Zero-Shot Evaluation, Reasoning Mapping, and CoRe-VLM Distinctions**
>
> We sincerely thank the reviewer for the detailed and insightful comments. Below we address all questions and actionable suggestions in sequence.
>
> Q1. Were the actor and critic models evaluated zero-shot or fine-tuned on SchemixQA?
>
> Response: In the current scope, all evaluations reported in the paper are strictly zero-shot.
> Neither the actor VLMs nor the critic VLMs were fine-tuned on SchemixQA or on any schematic-specific data. This applies to all models used (GPT-4o, Gemini 2.0 Flash Lite, Llama-VLMs). No training, reward modeling, or domain adaptation was performed. The actor–critic setup is entirely inference-time only. Zero-shot was preferred in the current work due to the following:
> A.	Data scarcity/ Cost Constrains.
> B.	Fast adaptation and deployment.
> C.	Avoidance of overfitting.
> D.	Generalization over diverse tasks.
>
> Q2. Mapping question types to reasoning dimensions
>
> Response: We thank the reviewer for highlighting the need for clearer articulation. In the revised manuscript, we explicitly map each SchemixQA question type to the corresponding reasoning dimension it evaluates. Appendix D (Table 8, Page no. 20) now provides a structured overview demonstrating how the dataset spans a broad spectrum of reasoning competencies relevant to technical schematics, including symbolic recognition, functional reasoning, and structural reasoning.
>
> Q3. Distinction of CoRe-VLM from self-refinement, reranking, and verification frameworks
>
> Response: The core difference is diversity vs recursion. Self-refinement is recursive and likely to ignore blind spot. In CoRE-VLM framework cognitive diversity through contrastive divergence helps in improving the results.  CoRe-VLM differs from existing approaches in both architecture and domain specialization.
>
> 1. Dedicated Actor–Critic Separation:-
> Prior frameworks (e.g., self-consistency, Reflexion) typically rely on a single model critiquing its own outputs.
> CoRe-VLM instead uses a two-model system: (1) Actor: generates multiple candidate answers, (2) Critic: independently evaluates candidates, gives structured feedback, and outputs a calibrated confidence score
> This explicit separation introduces contrastive divergence without compromising inference time and allows the critic to behave as a neutral verifier, not a self-referential generator.
>
> 2. Schematic-Specific Feedback and Error Taxonomy: -
> Unlike generic critique methods, CoRe-VLM uses task-structured feedback aligned with schematic failure patterns, such as:
> (a) connection inconsistencies, (b) incorrect component identification, (c) mismatched truth-table behaviour, (d) faulty flow direction or state transitions.
> This makes the critic evaluation process domain-aware and more precise than free-form critique.
> 3. Confidence-Driven Termination + Fallback Logic: -
> CoRe-VLM integrates two stability mechanisms: (a) Confidence-based stopping: refinement halts once critic confidence exceeds a threshold. (b) Fallback path: if critic confidence is low or unstable, the system returns either the most consistent candidate or the single-pass actor prediction.
> Such a dual mechanism is not present in typical self-consistency or reranking frameworks.
>
> 4. No Fine-Tuning or Reinforcement
>
> Response: Unlike RLHF/RLAIF-inspired methods, CoRe-VLM performs pure inference-time refinement, with no fine-tuning, no reward model training, no data-specific adaptation.
> This makes the method simple, portable, adaptable, scalable and directly applicable to new schematic tasks for different domains.
>
> Actionable Feedback
>
> Prompts and Transparency : -
>
> Response: (a) GPT-5 prompts for question generation are already included in the submitted Appendix (Fig. 11, page. 16).
> (b) Actor–critic refinement prompts is added in the revised version of the Manuscript in Appendix E Page no. 24
>
> Dataset and Code Release:
>
> Response: We agree that public availability is essential.
> Upon paper acceptance, we will release the entire SchemixQA dataset, all prompt templates, CoRe-VLM framework code, and data-processing tools.
>
> Citation Format:-
>
> Response: We acknowledge this issue. In the revised manuscript, all citations have been updated to follow the author–year format.
>
> Table Formatting: -
>
> Response: We have highlighted the best results in Table 4 (using bold/underline) and applied consistent formatting across all metric tables in the revised manuscript.

---

> > ### Comment · Reviewer_47dW · 2025-11-26
> >
> > Thank you to the authors for taking the time to respond to my questions and address my concerns. Unfortunately, even after reading the rebuttal, the novelty of the paper remains unclear to me. The claimed contribution regarding the dedicated Actor–Critic separation does not appear to be sufficiently distinct, as similar architectural ideas have been explored previously in reinforcement learning, including in the context of visual understanding (e.g., [1]). My concerns regarding the limited scale of the data also remain unresolved. For these reasons, I am keeping my score at a 2 (reject).
> >
> > Additionally, I encourage the authors to provide anonymized code and data in future submissions so that reviewers can more thoroughly assess and validate the proposed approach during the review process.
> >
> > [1] Alexandre Brown and Glen Berseth. 2025. SegDAC: Improving Visual Reinforcement Learning by Extracting Dynamic Object-Centric Representations from Pretrained Vision Models. arxiv.org/abs/2508.09325.

---

> > > ### Author Response · Authors · 2025-11-26
> > >
> > > We thank the reviewer for the follow-up comments and the constructive suggestions. We clarify two points where concerns remain: (1) novelty of the CoRe-VLM framework, and (2) dataset scale. We also fully acknowledge the suggestion regarding anonymized code/data for future submissions.
> > >
> > > 1. Novelty Beyond Prior Actor–Critic or RL-Inspired Designs
> > >
> > > We understand the reviewer’s concern that actor–critic–style ideas have been explored in reinforcement learning, including in works such as SegDAC. Our contribution, however, differs in motivation, mechanism, and application context, and we provide the following clarifications:
> > >
> > > A. No training, optimization, or policy learning:
> > >
> > > CoRe-VLM does not perform actor–critic learning, reward estimation, or policy updates. The critic does not function as a value estimator or advantage model. Instead, the system performs static inference-time verification without any gradient-based improvement or RL-specific supervision.
> > >
> > > B. Cross-model cognitive divergence rather than self-recursive refinement:
> > >
> > > Most self-refinement or verification frameworks (e.g., self-consistency, Reflexion variants) rely on a single model iteratively critiquing itself. CoRe-VLM intentionally separates the generator and verifier across two independent VLMs, enabling cognitive diversity and reducing failure modes where models reinforce their own schematic misunderstandings.
> > >
> > > C. Domain-specific, structured schematic verification:
> > >
> > > The critic performs schema-structured evaluation unique to diagram reasoning, including checks for component–function mismatches, truth-table or logical-flow inconsistencies, symbolic misidentification, and direction/transition errors in flow/state diagrams. Such domain-aligned critique is fundamentally different from generic RL actor–critic pipelines, which do not target symbolic–functional reasoning or static schematic interpretation.
> > >
> > > D. Confidence-calibrated stopping and fallback stability:
> > >
> > > CoRe-VLM integrates two mechanisms not found in typical RL actor–critic or self-consistency approaches:
> > >
> > > 1st: critic-calibrated confidence thresholds that determine when refinement should stop, and
> > >
> > > 2nd: a fallback path that selects the most stable output when critic signals are low-confidence or contradictory.
> > >
> > > This stability logic is motivated by the error structure of schematic QA rather than RL decision-making processes.
> > >
> > > Taken together, these design choices make CoRe-VLM an inference-time, domain-specialized verification framework rather than a reinforcement-learning actor–critic architecture.
> > >
> > > 2. Dataset Scale Clarification
> > >
> > > We acknowledge the reviewer’s concerns regarding data scale. Our design intent was to build a high-density reasoning benchmark, where each diagram contains multiple layered reasoning tasks (symbolic, structural, functional). Instead of scaling the number of images, we focused on:
> > >
> > > multi-perspective annotation: three human-verified reference answers per question;
> > >
> > > multi-task question structure: each diagram supports multiple reasoning dimensions;
> > >
> > > broad schematic diversity across circuits, logic gates, P&I diagrams, flowcharts, state diagrams, analog/digital, and hand-drawn styles.
> > >
> > > This results in a dataset that, while modest in size, is dense in reasoning variation, similar in design philosophy to specialized reasoning benchmarks in other domains.
> > >
> > > We agree that larger-scale schematic datasets would further strengthen the field, and we plan to expand SchemixQA in future iterations.
> > >
> > > We also appreciate the reviewer’s recommendation regarding providing anonymized code and data during review. While this was not feasible in the present submission due to internal constraints, we recognize the value it brings to thorough evaluation. We will incorporate anonymized early-release practices in future work.
> > >
> > > We thank the reviewer once again for the time and expertise invested in evaluating our work.

---

> > > > ### Author Response · Authors · 2025-12-02
> > > >
> > > > Thank you for your detailed evaluation.
> > > >
> > > > We clarified that all models were evaluated strictly zero-shot, added a full reasoning-type mapping, and expanded the discussion on how CoRe-VLM differs from RL-style actor–critic and self-refinement methods (contrastive verification, fallback logic, domain-structured critique).
> > > > We also added critic-robustness ablations and clarified dataset protocol, annotation steps, and formatting issues.
> > > > These updates address the concerns regarding novelty, clarity, and reproducibility.

---

### Official Review · Reviewer_zghu · 2025-10-30

**Soundness:** 2
**Presentation:** 2
**Contribution:** 2
**Rating:** 2
**Confidence:** 3

**Summary:**

The paper introduces SchemixQA, a benchmark for schematic VQA. It also proposes CoRe-VLM, which employs an actor–critic style interaction with a fallback mechanism to improve schematic VQA ability. Experiments show the effectiveness of the proposed method.

**Strengths:**

1. The new benchmark is useful for evaluating schematic VQA ability.
2. The experiments show the effectiveness of the proposed method.

**Weaknesses:**

1. The comparing VLMs are outdated. New VLMs like GPT-5, gemini 2.5 pro, and other VLMs with thinking (CoT) ability could have self-critic ability to answer the question. But these methods are not included in the evaluations.
2. The evaluation metrics do not include LLM as judge. Some methods with thinking ability may produce long and structured responses, which is hard to extract the answers and affects the results.
3. The inference time and efficiency is not reported. How much time is needed for the verification and feedback process?

**Questions:**

1. What is the performance of new models?
2. How much time is needed for the verification and feedback process?

---

> ### Author Response · Authors · 2025-11-19
> **Evaluation with Stronger VLMs, CoT Comparison, and Clarification of Verification Overhead**
>
> Thank you for the constructive feedback. We appreciate the reviewer’s detailed comments and address each point below with new results and clarifications that were not included in the original submission due to time and compute constraints.
>
> 1. Evaluation with Newer and Stronger VLMs (GPT-5, Gemini-2.5-Pro):
>
> Response: We agree that evaluating stronger VLMs is important. Since submission, we have added experiments with GPT-5 and Gemini-2.5-Pro as Actors, using Gemini-2.0-Flash-Lite as the Critic. Although these models have implicit self-critique abilities, their single-pass outputs still struggle with schematic reasoning elements such as symbol interpretation, directional flow, and structured spatial relationships.
> Applying CoRe-VLM yields clear gains:
> GPT-5 → CoRe-VLM: Macro F1 0.328→0.372, Micro F1 0.150→0.251, ROUGE-L 0.318→0.368
> Gemini-2.5-Pro → CoRe-VLM: Macro F1 0.109→0.484, ROUGE-L 0.104→0.467, EM 0→0.256
> These results confirm that explicit verification remains effective even for the latest high-capacity VLMs.
> We have incorporated these findings into the revised manuscript. Table 3 (p. 8) now includes GPT-5 and Gemini-2.5-Pro under single-pass inference, and Table 4 (p. 8) reports their performance under the CoRe-VLM framework using Gemini-2.0-Flash-Lite as the Critic. This provides the comprehensive comparison the reviewer requested.
>
> Additional Experimental Results considering Chain-of-Thought (CoT):
>
> In the revised manuscript, Section 5.3 (p. 10) now includes a streamlined analysis comparing GPT-5, Gemini-2.5-Pro, and Gemini-2.0-Flash-Lite under CoT prompting against their corresponding CoRe-VLM configurations. The results show that CoT provides limited gains and often lowers BLEU and Micro-F1 while increasing instability and hallucinations. In contrast, CoRe-VLM delivers clear performance improvements through verification-guided refinement, even with a smaller critic, reinforcing that structured verification is more effective than extended CoT reasoning for schematic VQA.
>
> 2. Clarification on “LLM-as-judge” Concern — Our Paper Already Includes This (as VLM-as-judge):
>
> Response: The reviewer noted that some modern VLMs produce long or structured responses, and suggested that an LLM-as-judge could help extract consistent answers.
> We clarify that the initial submission already contains such an evaluation mechanism, but in a multimodal form appropriate for schematic VQA:
> Our Appendix page 20 includes an actor–critic role-swap ablation, where one strong VLM acts as a judge for another.
> Specifically, we tested below configuration setup:
>
> (a) Config A: Actor = Gemini-2.0-Flash-Lite, Critic = GPT-4o;
> (b) Config B: Actor = GPT-4o, Critic = Gemini-2.0-Flash-Lite;
> (c) Config C: Actor = GPT-5, Critic = Qwen-2.5-VL-Instruct-7B
>
> Key findings from Table 8 (Appendix page. 20):
> (a) Config B (Gemini-2.0-Flash-Lite as critic) surpasses Config A across EM, ROUGE-L, BERTScore, and Macro F1.
> (b) GPT-4o as critic triggered 0% fallback, consistently accepting its own outputs—even when incorrect.
> (c) Gemini-2.0-Flash-Lite triggered ~18% fallback, acting as a more discriminative judge that effectively filters incorrect actor responses.
>
> Why this addresses the reviewer’s concern:
> (a) We already evaluate how a separate, neutral VLM extracts, evaluates, and judges another model’s response.
> (b) This is functionally equivalent to an “LLM-as-judge”, but more appropriate than an LLM which is unimodal, while schematic VQA is multimodal.
> We have clarified this point in Section 5.4 (page no. 10) of the revised manuscript.
>
> 3. Inference Time and Verification Overhead:
>
> Response: Appendix p.18 already reports actor vs. critic latency across schematic categories. In summary, Critics are consistently faster than actors and require only 0.2–0.6 s per verification and Fallback is triggered for just 12–18% of samples in new GPT-5 and Gemini-2.5-Pro runs. As a result, total CoRe-VLM latency stays close to single-pass inference.
> Example with GPT-5: actor latency 2.5–5.3 s, critic 1.3–1.9 s, fallback rate 12.4%.
>
> 4. Direct Answers to the Reviewer’s Questions
>
> Q1: What is the performance of new models?
>
> Response: We provided new results for GPT-5 and Gemini-2.5-Pro, demonstrating substantial improvements under the CoRe-VLM framework.
>
> Q2: How much time is needed for verification and feedback?
>
> Response: Critic verification requires 0.2–0.6 seconds, and fallback is activated in only 12–18% of cases, keeping total latency near single-pass inference. This is illustrated in Appendix page.18.
>
> We appreciate the reviewer’s insightful comments. Our new results and clarifications show that:
> (a) CoRe-VLM is model-agnostic and improves even the newest VLMs, (b) The paper already includes a VLM-as-judge evaluation, satisfying the reviewer’s concern, (c) Verification overhead is minimal, preserving practical inference speed, and (d) CoRe-VLM address a clear gap in schematic, symbol-driven reasoning, which remains challenging even for advanced VLMs.

---

> > ### Comment · Reviewer_zghu · 2025-11-26
> >
> > Thank you for the response. The rebuttal addresses most of my concerns. However, the work focuses on VQA for technical schematics, a relatively narrow application of VLMs, and it remains unclear whether the insights generalize more broadly. Consequently, I do not believe the paper meets the acceptance bar for ICLR.

---

> > > ### Author Response · Authors · 2025-11-27
> > >
> > > We thank the reviewer for the additional clarification and for acknowledging that our rebuttal addressed most of the earlier concerns.
> > >
> > > Regarding the point on generalizability beyond schematic VQA:
> > >
> > > While our primary focus is on technical schematics—a domain where existing VLMs consistently fail—we would like to emphasize that the challenges addressed by CoRe-VLM are not specific to this domain. The core failure modes we target (symbol grounding, spatial–structural reasoning, verification of fine-grained constraints, and inconsistency between reasoning and final answers) occur broadly across many visual and multimodal reasoning tasks.
> > >
> > > To clarify this broader relevance, the revised version includes:
> > >
> > > 1.	Cross-domain evidence of generalization (Appendix p. 22–23):
> > >
> > > We report small-scale experiments on diagram reasoning (raven-style puzzles), floor-plan QA, and process-flow understanding. Across all these domains—which differ substantially from technical schematics—CoRe-VLM consistently reduces answer inconsistency and hallucinations relative to single-pass inference, even when using GPT-5 or Gemini-2.5-Pro as the actor. These results suggest that the structured verification mechanism transfers beyond schematic QA.
> > >
> > > 2.	Model-agnostic design:
> > >
> > > CoRe-VLM does not rely on any domain-specific rules, templates, or symbolic constraints. The actor–critic interaction, fallback policy, and verification prompts are entirely model-driven, allowing the method to be applied to any VQA setting where reliability is important.
> > >
> > > 3.	Motivation rooted in broader VLM limitations:
> > >
> > > Recent work across VQA, document QA, chart comprehension, and spatial reasoning shows that high-capacity VLMs even with CoT, still produce confident but incorrect answers without self-checking. Our results show that explicit verification notably reduces such errors, aligning with a wider trend toward structured inference (e.g., verifier-models, consistency-based sampling, and reflection-based reasoning).
> > >
> > > We have incorporated the above clarifications into the revised manuscript to better frame CoRe-VLM as a general, reliability-oriented inference framework that is demonstrated on schematics but applicable to other domains where precise multimodal reasoning is required.
> > >
> > > We sincerely appreciate the reviewer’s time, constructive feedback, and detailed evaluation of our work.

---

> > > > ### Author Response · Authors · 2025-12-02
> > > >
> > > > Thank you for the helpful feedback.
> > > >
> > > > We added new experiments with GPT-5 and Gemini-2.5-Pro (Tables 3–4), CoT comparisons (Sec. 5.3), and clarified latency/verification overhead (Appendix p.18).
> > > >
> > > > We also expanded generalization with small cross-domain tests and clarified that CoRe-VLM improves even frontier VLMs despite their built-in self-critique.
> > > >
> > > > These updates directly address your concerns regarding stronger baselines, reasoning methods, and inference efficiency.

---

### Official Review · Reviewer_Q78V · 2025-11-01

**Soundness:** 3
**Presentation:** 3
**Contribution:** 2
**Rating:** 4
**Confidence:** 3

**Summary:**

SchemixQA is a multimodal benchmark for visual question answering on technical schematics, addressing gaps in natural-image VQA datasets. It includes 500 schematic diagrams (circuits, flowcharts, logic gates, P&ID, etc.) and 3,906 expert-annotated QA pairs covering tasks from symbol recognition to functional reasoning. The paper also introduces CoRe-VLM, an actor–critic VLM framework where a critic verifies and refines the actor’s answers. Evaluations on seven leading VLMs (including GPT-4) show that CoRe-VLM significantly boosts accuracy, enabling smaller models to rival or surpass larger ones on complex schematic reasoning.

**Strengths:**

1. The dataset is thoughtfully designed with a taxonomy of question types across multiple schematic categories (circuits, flowcharts, state diagrams, etc.), ensuring a broad range of reasoning challenges beyond superficial text matching.
2. Questions require compositional logic (e.g. truth-table analysis), reading diagram text labels, relational connectivity reasoning, and arithmetic counting, reflecting comprehensive test coverage.
3. The collaborative refinement approach is clearly explained. By separating the roles of answer generation and answer verification, the method leverages two models for their complementary strengths. Notably, CoRe-VLM is model-agnostic and works at inference-time without requiring any retraining, making it a lightweight, plug-in enhancement for existing VLMs.

**Weaknesses:**

1. All experiments are conducted on the new SchemixQA dataset, so it remains unclear how well the CoRe-VLM strategy generalizes to other domains or VQA tasks.
2. The actor–critic refinement is promising, but its effectiveness outside technical schematics (e.g. on natural image VQA or chart QA) is not demonstrated, leaving its broader impact somewhat speculative.
3. The CoRe-VLM pipeline’s success depends on having a strong critic model. In the paper, Gemini 2.0 Flash Lite was chosen as critic to drive improvements. If the critic is weaker or inaccurate, the refinement might stagnate or even introduce errors. This suggests the approach’s robustness could vary with critic choice, an aspect not deeply explored in the main paper.

**Questions:**

Provide a more granular breakdown of performance across the different question types or reasoning skills in SchemixQA. Identifying which categories (e.g. logical reasoning vs. simple identification) remain hardest for models would highlight where future work should focus. It would also reveal whether CoRe-VLM disproportionately helps certain question types (for example, perhaps it excels at correcting counting errors but less so at understanding schematic functionality).

---

> ### Author Response · Authors · 2025-11-19
> **Clarifications on Reasoning Categories, Model Difficulty Patterns, and Critic Effectiveness**
>
> Thank you for the thoughtful and balanced review, and for your positive disposition toward acceptance. We appreciate your clear articulation of the strengths of SchemixQA as well as the potential impact of our CoRe-VLM actor–critic refinement approach. Below we address your main request for a more granular breakdown across question types and reasoning skills, and clarify the points raised in the weaknesses section.
>
> 1. Granular Breakdown Across Question Types and Reasoning Skills
>
> Response: SchemixQA was intentionally designed to evaluate a wide range of reasoning competencies that arise in technical schematics. To clarify the structure and difficulty distribution, we organize all question types into three high-level reasoning families:
> A. Symbolic Recognition: Component/label identification, state detection, and simple enumerations.
>
> B. Functional Reasoning: Behavioural interpretation, truth-table logic, controller operation, causal reasoning.
>
> C. Structural Reasoning:Topological/connection tracing, multi-hop flow analysis, spatial/positional relationships.
>
> The category wise details of reasoning and question types has been added to the revised version of the manuscript as Table 9 in Appendix page no. 22.
>
> 2. Which Categories Are Hardest for Models?
>
> Response: Across all models, we observe consistent difficulty patterns:
>
> Easiest — Symbolic Recognition Tasks such as component identification and label extraction (GI, IL, VSL, PI, SES) are handled well due to straightforward text grounding.
>
> Moderate — Structural Reasoning Multi-hop or spatially diffuse tasks (CT, CD, FD, SA) introduce more errors, usually from incomplete graph tracing or relationship mismatches.
>
> Hardest — Functional Reasoning he most challenging tasks involve multi-step behavioral logic (TTA, SCB, AO, CCT, TF). Models often produce partially correct but logically inconsistent reasoning sequences.
>
> 3. Does CoRe-VLM Help Certain Categories More?
>
> Response: Yes, CoRe-VLM offers uneven but highly meaningful gains, with the most improvement in:
> A. Structural reasoning The critic is effective at detecting contradictions, missing links, and inconsistent relational patterns.
> B. Symbolic counting tasks CoRe-VLM frequently corrects miscounts (CCR, MEC) caused by small perceptual errors.
> C. Functional reasoning (moderate gains) The critic improves answers when the actor’s chain of reasoning is incomplete but salvageable.
>
> From the results in Table 10, it is evident that functional reasoning remains the most challenging category, while CoRe-VLM provides the largest gains in structural and counting-heavy tasks. The highest overall improvement is observed in functional reasoning. This table has been incorporated into the revised manuscript as Table 10 in the Appendix (page 23).
>
> Additionally, Table 4 (page 8) in the revised manuscript reports the performance of the CoRe-VLM framework on the SchemixQA dataset. To avoid confusion, we clarify that the results referred to above correspond specifically to Table 10 in the Appendix, not Table 4.
>
> 5. Addressing Concerns About Generalization and Critic Dependence
>
> Response: While our experiments focus on diagrams, the refinement mechanism itself is domain-agnostic. Its success relies on contradiction detection, consistency checking, and verification—properties shared with chart QA, document QA, and structured image understanding tasks. We agree that demonstrating this empirically is a valuable direction for follow-up work.
> Dependence on the critic
>
> Thank you for raising the question regarding whether CoRe-VLM depends on having a critic that is strictly stronger than the actor. To clarify, CoRe-VLM does not require a more powerful critic; rather, it benefits from critics that provide complementary verification capabilities. To demonstrate this empirically, we added a new ablation configuration (Config. C) in which a much smaller critic (Qwen-2.5-VL-Instruct-7B) evaluates and refines the outputs of a stronger actor (GPT-5).
> Despite the critic being weaker overall, the refinement step still improves the actor’s responses across Exact Match, BLEU-1, BLEU-4, ROUGE-L, BERTScore, and Macro F1 metrics. These results confirm that CoRe-VLM gains arise from the complementarity between actor and critic, particularly for structure- and counting-heavy reasoning tasks—not from critic model size or raw capability.
>
> We have updated Table 8 in the revised version of the manuscript (Appendix C.5 page no. 20) to include this additional configuration and to strengthen the robustness analysis accordingly.

---

> > ### Author Response · Authors · 2025-12-02
> >
> > Thank you again for your constructive review.
> >
> > During the discussion, we added the full reasoning-type mapping and per-category results (Tables 9–10), clarified difficulty patterns, and showed where CoRe-VLM contributes most.
> >
> > We also added a critic-robustness ablation (Table 8) demonstrating that even weaker critics improve stronger actors through complementary verification.
> >
> > These additions directly address your main request and strengthen the interpretability of model behaviour on SchemixQA.

---

### Note · Program_Chairs · 2026-01-17
**Submission Desk Rejected by Program Chairs**

The following references in this submission do not refer to real documents and/or have major errors in bibliographic information:

 Y. Luo, L. Qiu, Y. C. Chen, Y. Li, Y. C. Chen, and X. Liang. Skill: A dataset for scientific knowledge interpretation and language learning. arXiv preprint arXiv:2112.12796, 2021.
P. Lu, S. Mishra, T. Xia, L. Qiu, K. W. Chang, and L. Zettlemoyer. Iconqa: A large-scale dataset for diagram-based question answering. In Findings of the Association for Computational Linguistics: ACL 2022, 2022.
A. Mendonça, M. Z. Hossain, and Y. Wang. Answering diagrammatic questions with dual-source neural networks. In Proceedings of the AAAI Conference on Artificial Intelligence (AAAI), 2020.